# AN EFFICIENT SDE SOLVER FOR SCORE-BASED DIFFUSION MODELS

## ABSTRACT

Score-based (denoising diffusion) generative models have recently gained a lot of success in generating realistic and diverse data. These approaches define a forward diffusion process for transforming data to noise and generate data by reversing it (thereby going from noise to data). Unfortunately, current score-based models generate data very slowly due to the sheer number of score network evaluations required by numerical SDE solvers.

In this work, we aim to accelerate this process by devising a more efficient SDE solver. Existing approaches rely on the Euler-Maruyama (EM) solver, which uses a fixed step size. We found that naively replacing it with other SDE solvers fares poorly - they either result in low-quality samples or become slower than EM. To get around this issue, we carefully devise an SDE solver with adaptive step sizes tailored to score-based generative models piece by piece. Our solver requires only two score function evaluations per step, rarely rejects samples, and leads to high-quality samples. Our approach generates data 2 to 10 times faster than EM while achieving better or equal sample quality. For high-resolution images, our method leads to significantly higher quality samples than all other methods tested. Our SDE solver has the benefit of requiring no step size tuning.

## 1 INTRODUCTION

Score-based generative models (Song and Ermon, 2019; 2020; Ho et al., 2020; Jolicoeur-Martineau et al., 2020; Song et al., 2020a; Piché-Taillefer, 2021) have been very successful at generating data from various modalities, such as images (Ho et al., 2020; Song et al., 2020a), audio (Chen et al., 2020; Kong et al., 2020; Mittal et al., 2021; Kameoka et al., 2020), and graphs (Niu et al., 2020). They have further been used effectively for super-resolution (Saharia et al., 2021; Kadkhodaie and Simoncelli, 2020), inpainting (Kadkhodaie and Simoncelli, 2020), source separation (Jayaram and Thickstun, 2020), and image-to-image translation (Sasaki et al., 2021). In most of these applications, score-based models achieved superior performances in terms of quality and diversity than the historically dominant Generative Adversarial Networks (GANs) (Goodfellow et al., 2014).

Score-based models can be understood in two main classes: those based on a Variance Exploding (VE) diffusion process (Song and Ermon, 2019) and those based on a Variance Preserving (VP) one (Ho et al., 2020). Both diffusion processes progressively transform real data into Gaussian noise; $\mathcal{N}(\mathbf{0}, \sigma_{max}^2 \mathbf{I})$ for VE where $\sigma_{max}^2$ is very large, and $\mathcal{N}(\mathbf{0}, \mathbf{I})$ for VP.

The diffusion process (VE, VP, etc.) is then reversed in order to generate real data from Gaussian noise. Reversing the process requires the score function, which is estimated with a neural network (known as a score network). Although very powerful, score-based models generate data through an undesirably long iterative process; meanwhile, other state-of-the-art methods such as GANs generate data from a single forward pass of a neural network. Increasing the speed of the generative process is thus an active area of research.

Chen et al. (2020) and San-Roman et al. (2021) proposed faster step size schedules for VP diffusions that still yield relatively good quality/diversity metrics. Although fast, these schedules are arbitrary, require careful tuning, and the optimal schedules will vary from one model to another.

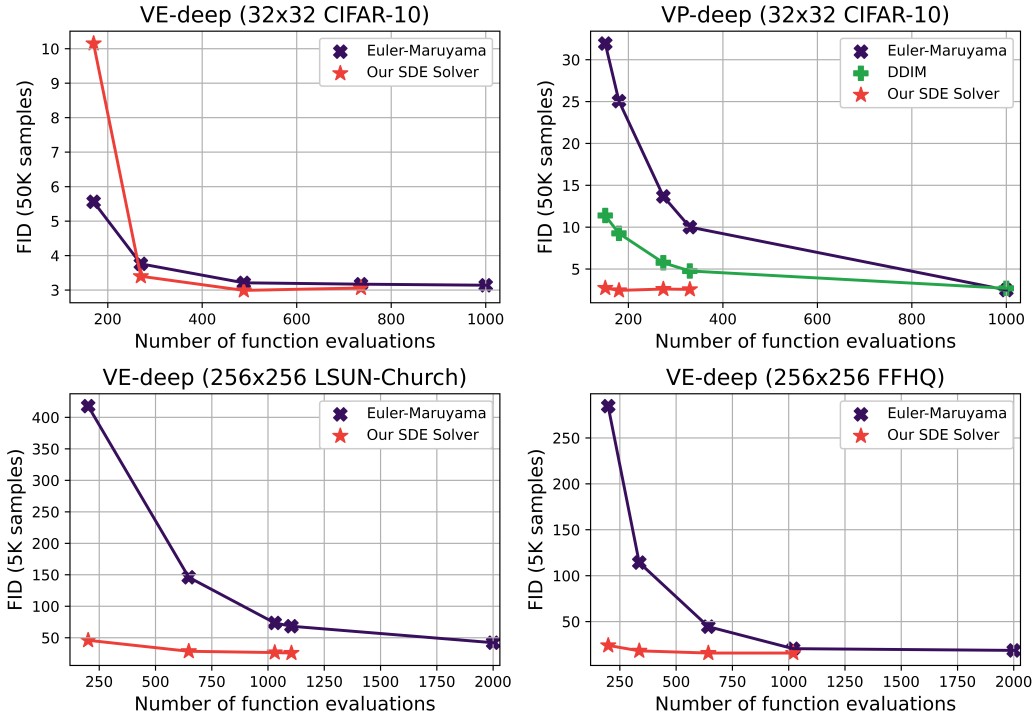

Figure 1: Comparison between our novel SDE solver at various values of error tolerance and Euler-Maruyama for an equal computational budget. We measure speed through the Number of Function Evaluations (NFE) and the quality of the generated images through the Fréchet Inception Distance (FID; lower is better). See Table 1-2 for more details.

Block et al. (2020) proposed generating data progressively from low to high-resolution images and show that the scheme improves speed. Similarly, Nichol and Dhariwal (2021) proposed generating low-resolution images and then upscale them since generating low-resolution images is quicker. They further suggested to accelerate VP-based models by learning dimension-specific noise rather than assuming equal noise everywhere. Note that these methods do not affect the data generation algorithm and would thus be complementary to our methods.

Song et al. (2020a) and Song et al. (2020b) proposed removing the noise from the data generation algorithm and solve an Ordinary Differential Equation (ODE) rather than a Stochastic Differential Equation (SDE); they report being able to converge much faster when there is no noise. Although it improves the generation speed, Song et al. (2020a) report obtaining lower-quality images when using the ODE formulation for the VE process (Song et al., 2020a). We will later show that our SDE solver generally leads to better results than ODE solvers at similar speeds.

Thus, existing methods for acceleration often require considerable step size/schedule tuning (this is also true for the baseline approach) and do not always work for both VE and VP processes. To improve speed and remove the need for step size/schedule tuning, we propose to solve the reverse diffusion process using SDE solvers with adaptive step sizes.

It turns out that off-the-shelf SDE solvers are ill-suited for generative modeling and exhibit either (1) divergence, (2) slower data generation than the baseline, or (3) significantly worse quality than the baseline (see Appendix A). This can be attributed to distinct features of the SDEs that arise in score-based generative models that set them apart from the SDEs traditionally considered in the numerical SDE solver literature, namely: (1) the codomain of the unknown function is extremely high-dimensional, especially in the case of image generation; (2) evaluating the score function is computationally expensive, requiring a forward pass of a large mini-batch through a large neural network; (3) the required precision of the solution is smaller than usual because we are satisfied as long as the error is not perceptible (e.g., one RGB increment on an image).

We devise our own SDE solver with these features in mind, resulting in an algorithm that can get around the problems encountered by off-the-shelf solvers. To address high dimensionality, we use the $\ell_2$ norm rather than the $\ell_\infty$ norm to measure the error across different dimensions to prevent a single pixel from slowing down the solver. To address the cost of score function evaluations while still obtaining high precision, we (1) take the minimum number of score function evaluations needed for adaptive step sizes (two evaluations), and (2) use extrapolation to get high precision at no extra cost. To take advantage of the reduced requirement for precision, we set the absolute tolerance for the error according to the range of RGB values.

Our main contribution is a new SDE solver tailored to score-based generative models with the following benefits:

- Our solver is much faster than the baseline methods, i.e. reverse-diffusion method with Langevin dynamics and Euler-Maruyama (EM);
- It yields higher quality/diversity samples than EM when using the same computing budget;
- It does not require any step size or schedule tuning;
- It can be used to quickly solve any type of diffusion process (e.g., VE, VP)

## 2 BACKGROUND

### 2.1 SCORE-BASED MODELING WITH SDES

Let $\mathbf{x}(0) \in \mathbb{R}^d$ be a sample from the data distribution $p_{\text{data}}$. The sample is gradually corrupted over time through a Forward Diffusion Process (FDP), a common type of Stochastic Differential Equation (SDE):

$$\mathrm{d}\mathbf{x} = f(\mathbf{x}, t)\mathrm{d}t + g(t)\mathrm{d}\mathbf{w}, \tag{1}$$

where $f(\mathbf{x}, t) : \mathbb{R}^d \times \mathbb{R} \to \mathbb{R}^d$ is the drift, $g(t) : \mathbb{R} \to \mathbb{R}$ is the diffusion coefficient and $\mathbf{w}(t)$ is the Wiener process indexed by $t \in [0, 1]$. Data points and their probability distribution evolve along the trajectories $\{\mathbf{x}(t)\}_{t=0}^1$ and $\{p_t(\mathbf{x})\}_{t=0}^1$ respectively, with $p_0 \equiv p_{\text{data}}$. The functions $f$ and $g$ are chosen such that $\mathbf{x}(1)$ be approximately Gaussian and independent from $\mathbf{x}(0)$. Inference is achieved by reversing this diffusion, drawing $\mathbf{x}(1)$ from its Gaussian distribution and solving the Reverse Diffusion Process (RDP) equal to:

$$\mathrm{d}\mathbf{x} = \left[ f(\mathbf{x}, t) - g(t)^2 \nabla_{\mathbf{x}} \log p_t(\mathbf{x}) \right] \mathrm{d}t + g(t)\mathrm{d}\bar{\mathbf{w}}, \tag{2}$$

where $\nabla_{\mathbf{x}} \log p_t(\mathbf{x})$ is referred to as the score of the distribution at time $t$ (Hyvärinen, 2005) and $\bar{\mathbf{w}}(t)$ is the Wiener process in which time flows backward (Anderson, 1982).

One can observe from Equation 2 that the RDP requires knowledge of the score (or $p_t$), which we do not have access to. Fortunately, it can be estimated by a neural network (referred to as the score network) by optimizing the following objective:

$$\mathcal{L}(\theta) = \mathbb{E}_{\mathbf{x}(t) \sim p(\mathbf{x}(t)|\mathbf{x}(0)), \mathbf{x}(0) \sim p_{\text{data}}} \left[ \frac{\lambda(t)}{2} \left\| s_\theta(\mathbf{x}(t), t) - \nabla_{\mathbf{x}(t)} \log p_t(\mathbf{x}(t)|\mathbf{x}(0)) \right\|_2^2 \right], \tag{3}$$

where $\lambda(t) : \mathbb{R} \to \mathbb{R}$ is a weighting function generally chosen to be inversely proportional to:

$$\mathbb{E} \left[ \left\| \nabla_{\mathbf{x}(t)} \log p_t(\mathbf{x}(t)|\mathbf{x}(0)) \right\|_2^2 \right].$$

One can demonstrate that the minimizer of that objective $\theta^*$ will be such that $s_{\theta^*}(\mathbf{x}, t) = \nabla_{\mathbf{x}} \log p_t(\mathbf{x})$ (Vincent, 2011), allowing us to approximate the reverse diffusion process. As can be seen, evaluating the objective requires the ability to generate samples from the FDP at arbitrary times $t$. Thankfully, as long as the drift is affine (i.e., $f(\mathbf{x}, t) = \mathbf{A}\mathbf{x} + \mathbf{B}$), the transition kernel $p(\mathbf{x}(t)|\mathbf{x}(0))$ will always be normally distributed (Särkkä and Solin, 2019), which means that we can solve the forward diffusion in a single step. Furthermore, the score of the Gaussian transition kernel is trivial to compute, making the loss an inexpensive training objective.

There are two primary choices for the FDP in the literature, which we discuss below.

## 2.2 VARIANCE EXPLODING (VE) PROCESS

The Variance Exploding (VE) process consists in the following FDP:

$$\mathrm{d}\mathbf{x} = \sqrt{\frac{\mathrm{d}\left[\sigma^2(t)\right]}{\mathrm{d}t}}\mathrm{d}\mathbf{w}.$$

Its associated transition kernel is:

$$\mathbf{x}(t)|\mathbf{x}(0) \sim \mathcal{N}(\mathbf{x}(0), [\sigma^2(t) - \sigma^2(0)]\mathbf{I}) \approx \mathcal{N}(\mathbf{x}(0), \sigma^2(t)\mathbf{I}).$$

In practice, we let $\sigma(t) = \sigma_{min}\left(\frac{\sigma_{max}}{\sigma_{min}}\right)^t$, where $\sigma_{min} = 0.01$ and $\sigma_{max} \approx \max_i \sum_{j=1}^{N} ||\mathbf{x}^{(i)} - \mathbf{x}^{(j)}||$ is the maximum Euclidean distance between two samples from the dataset $\{\mathbf{x}^{(i)}\}_{i=1}^{N}$ (Song and Ermon, 2020). Using the maximum Euclidean distance ensures that $\mathbf{x}(1)$ does not depend on $\mathbf{x}(0)$; thus, $\mathbf{x}(1)$ is approximately distributed as $\mathcal{N}(\mathbf{0}, \sigma^2(1)\mathbf{I})$.

## 2.3 VARIANCE PRESERVING (VP) PROCESS

The Variance Preserving (VP) process consists in the following FDP:

$$\mathrm{d}\mathbf{x} = -\frac{1}{2}\beta(t)\mathbf{x}\mathrm{d}t + \sqrt{\beta(t)}\mathrm{d}\mathbf{w}.$$

Its associated transition kernel is:

$$\mathbf{x}(t)|\mathbf{x}(0) \sim \mathcal{N}\left(\mathbf{x}(0)\, e^{-\frac{1}{2}\int_0^t \beta(s)\mathrm{d}s}, \left(1 - e^{-\int_0^t \beta(s)\mathrm{d}s}\right)\mathbf{I}\right).$$

In practice, we let $\beta(t) = \beta_{min} + t\,(\beta_{max} - \beta_{min})$, where $\beta_{min} = 0.1$ and $\beta_{max} = 20$. Thus, $\mathbf{x}(1)$ is approximately distributed as $\mathcal{N}(\mathbf{0}, \mathbf{I})$ and does not depend on $\mathbf{x}(0)$.

## 2.4 SOLVING THE REVERSE DIFFUSION PROCESS (RDP)

There are many ways to solve the RDP; the most basic one being Euler-Maruyama (Kloeden and Platen, 1992), the SDE analog to Euler's method for solving ODEs. Song et al. (2020a) also proposed *Reverse-Diffusion*, which consists in ancestral sampling (Ho et al., 2020) with the same discretization used in the FDP. With the Reverse-Diffusion, (Song et al., 2020a) obtained poor results unless applying an additional Langevin dynamics step after each Reversion-Diffusion step. They named this approach Predictor-Corrector (PC) sampling, with the predictor corresponding to Reverse-Diffusion and the corrector to Langevin dynamics. Although using a corrector step leads to better samples, PC sampling is only heuristically motivated and the theoretical underpinnings are not yet understood. Nevertheless, (Song et al., 2020a) report their best results (in terms of lowest Fréchet Inception Distance (Heusel et al., 2017)) using the Reverse-Diffusion with Langevin dynamics for VE models. For VP models, they obtain their best results using Euler-Maruyama.

## 3 EFFICIENT METHOD FOR SOLVING REVERSE DIFFUSION PROCESSES

### 3.1 SETTING UP THE ALGORITHM

We start with a general algorithm for solving an SDE (similar to most ODE/SDE solvers). We choose the various options/hyper-parameters based on a mixture of theory and experiments; an ablation study of the different hyper-parameters can also be found in Appendix B.

### 3.1.1 INTEGRATION METHOD

Solving the RDP to generate data can take an undesirably long time. One would assume that solving SDEs with high-order methods would improve speed over Euler-Maruyama, just like high-order ODE solvers improve speed over Euler's method when solving ODEs. However, this is not always the case: while higher-order solvers may achieve lower discretization errors, they require more function

evaluations, and the improved precision might not be worth the increased computation cost (Lehn et al., 2002; Lamba, 2003).

Our preliminary attempts at using SDE solvers with the *DifferentialEquations.jl* Julia package (Rackauckas and Nie, 2017a) confirmed that higher-order methods were significantly slower (6 to 8 times slower; see Appendix A). Lamba's algorithm (Lamba, 2003), a low-order adaptive method, yielded the fastest results, thus motivating us to restrict our search to the space of low-order methods. Still, the resulting images were of lower quality.

Using a fixed step-size while solving an ODE/SDE requires some tuning and one should be able to advance faster (from $t = 1$ to $t = 0$) in regions of low noise. To gain more speed, one can dynamically adjust the step size over time; this is a common approach used in most fast ODE/SDE solvers. Such technique generally use two integration methods: a lower-order ($\mathbf{x}'$) method and a higher-order ($\mathbf{x}''$) method. The local error $E(\mathbf{x}', \mathbf{x}'') = \mathbf{x}' - \mathbf{x}''$ is used to determine how stable the lower-order method is at the current step size; the closer to zero, the more appropriate the step size is. From this information, we can dynamically adjust the step size and decide whether or not to accept the proposed sample of the lower-order method. Alternatively, one can select $\mathbf{x}''$ as the proposal, which we will refer to as *extrapolating*.

Rather than using the Improved Euler ODE solver (Süli and Mayers, 2003) as in Lamba (2003) or a high-order stochastic Runge-Kutta method (Rößler, 2010) as in Rackauckas and Nie (2017b) (which did not work well in our preliminary attempts with the Julia package) we instead rely on the more recent Improved Euler SDE solver (Roberts, 2012) as our higher order method. This method is very similar to the classical Improved Euler ODE solver, but it is made to work with SDEs instead of ODEs. Importantly, this method only requires two score function evaluations and re-uses the same score function evaluation used for EM, meaning that it is only twice as expensive as EM. Similarly to Lamba's algorithm, this method, albeit quick, leads to images of poor quality. However, by using extrapolation (taking $\mathbf{x}''$ instead of $\mathbf{x}'$ as our proposal), we were able to match and improve over the baseline approach (EM). Thus, using the stochastic Improved Euler was the key to taking bigger steps without sacrificing precision. Note that Lamba's algorithm cannot use extrapolation due to its use of a non-stochastic ODE solver (Improved Euler).

An algorithm has strong-order $p$ when the local error from $t$ to $t + h$ is $\mathcal{O}(h^{p+1})$). Euler-Maruyama has strong-order 0.5 while Improved Euler has strong-order 1 (Roberts, 2012). The highest strong-order found in the *DifferentialEquations.jl* Julia package (Rackauckas and Nie, 2017a) are order 1.5. Thus, our method obtains a balance between methods that are 1) low precision, but fast and 2) high precision, but slow.

### 3.1.2 TOLERANCE

In ODE/SDE solvers, the local error is divided by a *tolerance* term. Traditionally, the mixed tolerance $\boldsymbol{\delta}(\mathbf{x}') : \mathbb{R}^d \to \mathbb{R}^d$ is calculated as the maximum between the absolute and relative tolerance:

$$\boldsymbol{\delta}(\mathbf{x}') = \max(\epsilon_{abs}, \epsilon_{rel}|\mathbf{x}'|), \tag{4}$$

where the absolute value $|\cdot|$ is applied element-wise.

The *DifferentialEquations.jl* Julia package instead calculates the mixed tolerance through the maximum of the current and previous sample:

$$\boldsymbol{\delta}(\mathbf{x}', \mathbf{x}'_{prev}) = \max(\epsilon_{abs}, \epsilon_{rel} \max(|\mathbf{x}'|, |\mathbf{x}'_{prev}|)). \tag{5}$$

Having no trivial prior for which approach to use, we tried both and found the latter approach (Equation 5) to converge much faster for VE models (see Appendix B).

Given our focus on image generation, we can set $\epsilon_{abs}$ a priori. During training and at the end of the data generation, images are represented as floating-point tensors with range $[y_{min}, y_{max}]$. When evaluated, they must be transformed into 8-bit color images; this means that images are scaled to the range $[0, 255]$ and converted to the nearest integer (to represent one of the 256 values per color channel). Given the 8-bit color encoding, an absolute tolerance $\epsilon_{abs} = \frac{y_{max} - y_{min}}{256}$ corresponds to tolerating local errors of at most one color (e.g., $x'_{ij}$ with Red=5 and $x''_{ij}$ with Red=6 is accepted, but $x'_{ij}$ with Red=5 and $x''_{ij}$ with Red=7 is not) channel-wise. One-color differences are not perceptible and should not influence the metrics used for evaluating the generated images. For VP models, which

have range $[-1, 1]$, this corresponds to $\epsilon_{abs} = 0.0078$ while for VE models, which have range $[0, 1]$, this corresponds to $\epsilon_{abs} = 0.0039$.

To control speed/quality, we vary $\epsilon_{rel}$, where large values lead to more speed but less precision, while small values lead to the converse.

### 3.1.3   NORM OF THE SCALED ERROR

The scaled error (the error scaled by the mixed tolerance) is calculated as

$$E_q = \left\| \frac{\mathbf{x}' - \mathbf{x}''}{\boldsymbol{\delta}(\mathbf{x}', \mathbf{x}'_{prev})} \right\|_q.$$

Many algorithms use $q = \infty$ (Lamba, 2003; Rackauckas and Nie, 2017b), where $||\mathbf{x}||_\infty = \max(\mathbf{x}_1, ..., \mathbf{x}_k)$ over all $k$ elements of $\mathbf{x}$. Although this can work with low-dimensional SDEs, this is highly problematic for high-dimensional SDEs such as those in image-space. The reason is that a single channel of a single pixel (out of 65536 pixels for a $256 \times 256$ color image) with a large local error will cause the step size to be reduced for all pixels and possibly lead to a step size rejection. Indeed, our experiments confirmed that using $q = \infty$ slows down generation considerably (see Appendix B). To that effect, we instead use a scaled $\ell_2$ norm:

$$||\mathbf{x}||_2 = \sqrt{\frac{1}{n} \sum_{i=1}^{k} \left( \frac{\mathbf{x}' - \mathbf{x}''}{\boldsymbol{\delta}(\mathbf{x}', \mathbf{x}'_{prev})} \right)_k}.$$

### 3.1.4   HYPERPARAMETERS OF THE DYNAMIC STEP SIZE ALGORITHM

Upon calculating the scaled error, we accept the proposal $\mathbf{x}''$ if $E_q \leq 1$ and increment the time by $h$ Whether or not it is accepted, we update the next step size $h$ in the usual way:

$$h \leftarrow \min(h_{\max}, \theta h E_q^{-r}), \tag{6}$$

where $h_{\max}$ is the maximum step size, $\theta$ is the safety parameter which determines how strongly we adapt the step size (0 being very safe; 1 being fast, but high rejections rate), and $r$ is an exponent-scaling term.

Although ODE theory tells us that we should let $r = \frac{1}{p+1}$ with $p$ being the order of the lower-order integration method, there is no such theory for SDEs (Rackauckas and Nie, 2017b). Thus, as Rackauckas and Nie (2017b) suggest, we resorted to empirically testing values and found that any $r \in [0.5, 1]$ works well on both VE and VP processes, but that $r \in [0.8, 0.9]$ is slightly faster (see Appendix B). We arbitrarily chose $r = 0.9$ as the default setting.

Finally, we defaulted to setting $\theta = 0.9$ for the safety parameter as is common in the literature, and choose $h_{\max}$ to be equal to the largest step size possible, namely the remaining time $t$.

### 3.1.5   HANDLING THE MINI-BATCH

Using the same step size for every sample of a mini-batch means that every images would be slowed down by the other images. Since every image's RDP is independent from one another, we apply a different step size to each data sample; some images may converge faster than others, but we wait for all images to have converged.

### 3.2   ALGORITHM

In Section 3.1, We defined every aspect of the algorithm needed to numerically solve Equation 2 for image generation. The algorithm thus consists in using adaptive step sizes through Equation 6 with the hyperparameters defined in the previous subsection ($q = \infty$, $\theta = 0.9$, $r = 0.9$, $\epsilon_{abs} = \frac{y_{max} - y_{min}}{256}$) with Euler-Maruyama as the low-order method and Improved Euler as the high-order method. The resulting algorithm is described in Algorithm 1. This algorithm is straightforward to parallelize across the batch dimension.

Note that this algorithm is only for solving an RDP; a more general version for solving an arbitrary forward-time diffusion process can be found in Appendix C. Additionally, we present a demonstration in Appendix F showing that the extrapolation step conserves the stability and convergence of the EM step.

---

**Algorithm 1** Dynamic step size extrapolation for solving Reverse Diffusion Processes

---

**Require:** $s_\theta, \epsilon_{rel}, \epsilon_{abs}, h_{init} = 0.01, r = 0.9, \theta = 0.9$      ▷ for images: $\epsilon_{abs} = \frac{y_{max} - y_{min}}{256}$

  $t \leftarrow 1$
  $h \leftarrow h_{init}$
  Initialize $\mathbf{x}$
  $\mathbf{x}'_{prev} \leftarrow \mathbf{x}$
  **while** $t > 0$ **do**
    Draw $\mathbf{z} \sim \mathcal{N}(\mathbf{0}, \mathbf{I})$
    $\mathbf{x}' \leftarrow \mathbf{x} - hf(\mathbf{x}, t) + hg(t)^2 s_\theta(\mathbf{x}, t) + \sqrt{h}g(t)\mathbf{z}$      ▷ Euler-Maruyama
    $\tilde{\mathbf{x}} \leftarrow \mathbf{x} - hf(\mathbf{x}', t - h) + hg(t - h)^2 s_\theta(\mathbf{x}', t - h) + \sqrt{h}g(t - h)\mathbf{z}$
    $\mathbf{x}'' \leftarrow \frac{1}{2}(\mathbf{x}' + \tilde{\mathbf{x}})$      ▷ Improved Euler (SDE version)
    $\boldsymbol{\delta} \leftarrow \max(\epsilon_{abs}, \epsilon_{rel} \max(|\mathbf{x}'|, |\mathbf{x}'_{prev}|))$      ▷ Element-wise operations
    $E_2 \leftarrow \frac{1}{\sqrt{n}} \left\| (\mathbf{x}' - \mathbf{x}'') / \boldsymbol{\delta} \right\|_2$
    **if** $E_2 \leq 1$ **then**      ▷ Accept
      $\mathbf{x} \leftarrow \mathbf{x}''$      ▷ Extrapolation
      $t \leftarrow t - h$
      $\mathbf{x}'_{prev} \leftarrow \mathbf{x}'$
    $h \leftarrow \min(t, \theta h E_2^{-r})$      ▷ Dynamic step size update
  **return** $\mathbf{x}$

---

## 4 EXPERIMENTS

To test Algorithm 1 on RDPs, we apply it to various pre-trained models from Song et al. (2020a). To start, we generate low-resolution images (32x32) by testing the VP, VE, VP-deep, and VE-deep models on CIFAR-10 (Krizhevsky et al., 2009). Then, we generate higher-resolutions images (256x256) by testing the VE models on LSUN-Church (Yu et al., 2015), and Flickr-Faces-HQ (FFHQ) (Karras et al., 2019). See implementation details in Appendix D. We used four or less V100 GPUs to run the experiments.

To measure the performance of the image generation, we calculate the Fréchet Inception Distance (FID) (Heusel et al., 2017) and the Inception Score (IS) (Salimans et al., 2016), where low FID and high IS correspond to higher quality/diversity. We compare our method to the three base solvers used in Song et al. (2020a): Reverse-Diffusion with Langevin dynamics, Euler-Maruyama (EM), and Probability Flow, where the latter solves an ODE instead of an SDE using Runge-Kutta 45 (Dormand and Prince, 1980). We also compare against the fast solver by (Song et al., 2020b) called denoising diffusion implicit models (DDIM), which is only defined for VP models. We define the *baseline* approach as the solver used by Song et al. (2020a) which leads to the lowest FID (EM for VP models and Reverse-Diffusion with Langevin for VE models). For our algorithm, the only free hyperparameter is the relative tolerance which we set to $\epsilon_{rel} \in \{0.01, 0.02, 0.05, 0.1, 0.5\}$.

The FID and the Number of score Function Evaluations (NFE) are described in Table 1 for low-resolution images and Table 2 for high-resolution images. The Inception Score (IS) is described for CIFAR-10 in Appendix E.

### 4.1 PERFORMANCE

Compared to EM, we observe that our method is immediately advantageous in terms of quality/diversity for high-resolution images, along with 2 to 3× speedups ($\epsilon_{rel} = 0.02$). While this advantage becomes less obvious in terms of the FID on CIFAR-10, our method still offers $> 5\times$ computational speedups at no apparent disadvantage ($\epsilon_{rel} \in \{0.02, 0.05\}$).

Table 1: Number of score Function Evaluations (NFE) / Fréchet Inception Distance (FID) on CIFAR-10 (32x32) from 50K samples

| Method | VP | VP-deep | VE | VE-deep |
|---|---|---|---|---|
| Reverse-Diffusion & Langevin | 1999 / 3.41 | 1999 / 3.28 | 1999 / **2.40** | 1999 / **2.21** |
| Euler-Maruyama | 1000 / **2.55** | 1000 / **2.49** | 1000 / 2.98 | 1000 / 3.14 |
| DDIM | 1000 / 2.86 | 1000 / 2.69 | – | – |
| Ours ($\epsilon_{rel} = 0.01$) | 329 / 2.70 | 330 / 2.56 | 738 / 2.91 | 736 / 3.06 |
| Euler-Maruyama (same NFE) | 329 / 10.28 | 330 / 10.00 | 738 / 2.99 | 736 / 3.17 |
| DDIM (same NFE) | 329 / 4.81 | 330 / 4.76 | – | – |
| Ours ($\epsilon_{rel} = 0.02$) | 274 / 2.74 | 274 / 2.60 | 490 / **2.87** | 488 / **2.99** |
| Euler-Maruyama (same NFE) | 274 / 14.18 | 274 / 13.67 | 490 / 3.05 | 488 / 3.21 |
| DDIM (same NFE) | 274 / 5.75 | 274 / 5.74 | – | – |
| Ours ($\epsilon_{rel} = 0.05$) | 179 / **2.59** | 180 / **2.44** | 271 / 3.23 | 270 / 3.40 |
| Euler-Maruyama (same NFE) | 179 / 25.49 | 180 / 25.05 | 271 / 3.48 | 270 / 3.76 |
| DDIM (same NFE) | 179 / 9.20 | 180 / 9.25 | – | – |
| Ours ($\epsilon_{rel} = 0.10$) | 147 / 2.95 | 151 / 2.73 | 170 / 8.85 | 170 / 10.15 |
| Euler-Maruyama (same NFE) | 147 / 31.38 | 151 / 31.93 | 170 / 5.12 | 170 / 5.56 |
| DDIM (same NFE) | 147 / 11.53 | 151 / 11.38 | – | – |
| Ours ($\epsilon_{rel} = 0.50$) | 49 / 72.29 | 48 / 82.42 | 52 / 266.75 | 50 / 307.32 |
| Euler-Maruyama (same NFE) | 49 / 92.99 | 48 / 95.77 | 52 / 169.32 | 50 / 271.27 |
| DDIM (same NFE) | 49 / 37.24 | 48 / 38.71 | – | – |
| Probability Flow (ODE) | 142 / 3.11 | 145 / 2.86 | 183 / 7.64 | 181 / 5.53 |

Table 2: Number of score Function Evaluations (NFE) / Fréchet Inception Distance (FID) on LSUN-Church (256x256) and FFHQ (256x256) from 5K samples

| Method | VE (Church) | VE (FFHQ) |
|---|---|---|
| Reverse-Diffusion & Langevin | 3999 / 29.14 | 3999 / 16.42 |
| Euler-Maruyama | 2000 / 42.11 | 2000 / 18.57 |
| Ours ($\epsilon_{rel} = 0.01$) | 1104 / **25.67** | 1020 / **15.68** |
| Euler-Maruyama (same NFE) | 1104 / 68.24 | 1020 / 20.45 |
| Ours ($\epsilon_{rel} = 0.02$) | 1030 / **26.46** | 643 / **15.67** |
| Euler-Maruyama (same NFE) | 1030 / 73.47 | 643 / 44.42 |
| Ours ($\epsilon_{rel} = 0.05$) | 648 / 28.47 | 336 / 18.07 |
| Euler-Maruyama (same NFE) | 648 / 145.96 | 336 / 114.23 |
| Ours ($\epsilon_{rel} = 0.10$) | 201 / 45.92 | 198 / 24.02 |
| Euler-Maruyama (same NFE) | 201 / 417.77 | 198 / 284.61 |
| Probability Flow (ODE) | 434 / 214.47 | 369 / 135.50 |

Reverse-Diffusion with Langevin achieves the lowest FID for VE models on CIFAR-10, though at the cost of a $4\times$ computational overhead over our method. Furthermore, their advantage vanishes for VP models and when generating high-resolution images.

We further compare our SDE solver to EM given the same computational budget and observe that our method is always immensely preferable in high-resolutions and for VP models. For VE models on CIFAR-10, we observe that our algorithm leads to a better FID as long as the NFE is sufficiently large (270). Note that since our algorithm takes two score function evaluations per step, EM has the advantage of doing twice as many steps given the same NFE, which appears to be a factor more important than the order of the method at low budget in low-resolution VE. Nevertheless, comparing for equal number of iterative step, the results still point to our method being always preferable. For high-resolution images, we see that EM cannot converge on moderate to small NFEs due to the high-dimensionality, making of our SDE solver the way to go.

Generally, we observe that the VE process cannot be solved as fast as the VP process; this is due to the enormous Gaussian noise in the VE process causing larger local errors. This reflects the issue mentioned in Section 3.1.1 regarding high-order SDE solvers not always being beneficial in terms of speed for SDEs with heavy Gaussian noise. In practice, for VE, the algorithm uses a small step size in the beginning to ensure high accuracy and eventually increases the step size as the noise becomes less considerable.

## 4.2 SOLVING AN ODE INSTEAD OF AN SDE

We see that our SDE solver generally does better than Probability Flow, especially in high-resolution, where we obtain greatly lower FIDs with a similar budget. Our algorithm attains the same NFE as Probability Flow when $\epsilon_{rel} = 0.10$ for low-resolution images and when $0.05 < \epsilon_{rel} < 0.10$ for high-resolution images. For the same budget, Probability Flow has higher FID than our approach on all but low-resolution VE models. However, in that case, our algorithm achieves a much lower FID when $\epsilon_{rel} \leq 0.05$, albeit slower. In high-resolution, Probability Flow leads to very poor FIDs, suggesting no convergence.

## 4.3 DDIM

Contrary to Song et al. (2020b),the FID of DDIM worsens significantly when the NFE decreases. This could be due to differences between Song et al. (2020a) continuous-time score-matching and the DDIM training procedure and architecture. Nevertheless, the FID increase engendered by a reduced budget is much less dramatic than for EM. As of note, DDIM succeeds in maintaining a lower FID than our solver at extremely small NFEs ($< 50$), albeit with extremely poor FID.

## 5 LIMITATIONS

Although we tested our approach on a wide range of settings, we nevertheless only tested on continuous-time image generation models. We did so because solving the SDE requires continuous-time and the only such pre-trained models at time of publishing are the one by Song et al. (2020a).

Although our approach removes step size and schedule tuning, we still need to choose a value of the relative tolerance, which indirectly affects the number of steps taken; one could theoretically tune this hyper-parameter to optimize a certain metric, going against the point of removing tuning. Still, letting $\epsilon_{rel} = 0.01$ for precise results and $\epsilon_{rel} = 0.05$ for fast results are reasonable choices, as all evidence points to the FID being stable w.r.t. $\epsilon_{rel}$.

## 6 CONCLUSION

We built an SDE solver that allows for generating images of comparable (or better) quality to Euler-Maruyama at a much faster speed. Our approach makes image generation with score-based models more accessible by shrinking the required computational budgets by a factor of 2 to $5\times$, and presenting a sensible way of compromising quality for additional speed.

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

APPENDICES

## A  DIFFERENTIALEQUATIONS.JL

Table 3: Short experiments with various SDE solvers from *DifferentialEquations.jl* on the VP model with a small mini-batch.

| Method | Strong-Order | Adaptive | Speed |
|---|---|---|---|
| Euler-Maruyama (EM) | 0.5 | No | Baseline speed |
| SOSRA (Rößler, 2010) | 1.5 | Yes | 5.92 times **slower** |
| SRA3 (Rößler, 2010) | 1.5 | Yes | 6.93 times **slower** |
| Lamba EM (default) (Lamba, 2003) | 0.5 | Yes | Did not converge |
| Lamba EM (atol=1e-3) (Lamba, 2003) | 0.5 | Yes | 2 times **faster** |
| Lamba EM (atol=1e-3, rtol=1e-3) (Lamba, 2003) | 0.5 | Yes | 1.27 times **faster** |
| Euler-Heun | 0.5 | No | 1.86 times **slower** |
| Lamba Euler-Heun (Lamba, 2003) | 0.5 | Yes | 1.75 times **faster** |
| SOSRI (Rößler, 2010) | 1.5 | Yes | 8.57 times **slower** |
| RKMil (at various tolerances) (Kloeden and Platen, 1992) | 1.0 | Yes | Did not converge |
| ImplicitRKMil (Kloeden and Platen, 1992) | 1.0 | Yes | Did not converge |
| ISSEM | 0.5 | Yes | Did not converge |

Here, we report the preliminary experiments we ran with the *DifferentialEquations.jl* Julia package (Rackauckas and Nie, 2017a) before devising our own SDE solver. As can be seen, most methods either did not converge (with warnings of "instability detected") or converged, but were much slower than Euler-Maruyama. The only promising method was Lamba's method (Lamba, 2003). Note that an algorithm has strong-order $p$ when the local error from $t$ to $t + h$ is $\mathcal{O}(h^{p+1})$.

## B  EFFECTS OF MODIFYING ALGORITHM 1

Table 4: Effect of different settings on the [Inception score (IS) / Fréchet Inception Distance (FID) / Number of score Function Evaluations (NFE)] from 10k samples (with mini-batches of 1k samples) with the VP - CIFAR10 model.

| Change(s) in Algorithm 1 | IS | FID | NFE |
|---|---|---|---|
| No change $\left[q = 2, r = 0.9, \delta(\mathbf{x}', \mathbf{x}'_{prev})\right]$ | 9.38 | 4.70 | 3972 |
| Small modifications | | | |
| $\delta(\mathbf{x}')$ | 9.26 | 4.69 | 4166 |
| No Extrapolation (thus, using Euler–Maruyama) | 9.58 | 11.73 | 3978 |
| $q = \infty$ | 9.48 | 4.90 | 14462 |
| $r = .5$ | 9.41 | 4.69 | 4104 |
| $r = .8$ | 9.36 | 4.68 | 3938 |
| $r = 1$ | 9.41 | 4.69 | 4048 |
| Variations of Lamba (2003) Algorithm | | | |
| $r = 0.5$, Lamba integration | 7.80 | 52.98 | 1468 |
| $r = 0.5$, Lamba integration, Extrapolation | 7.32 | 64.65 | 1438 |
| $r = 0.5$, Lamba integration, $q = \infty$ | 9.28 | 21.09 | 2360 |
| $r = 0.5$, Lamba integration, $q = \infty, \theta = 0.8$ | 9.21 | 18.82 | 2346 |

As can be seen, most chosen settings lead to better results. However, $r$ seems to have little impact on the FID. Still, using $r \in [0.8, 0.9]$ lead to a little bit less score function evaluations and sometimes lead to lower FID.

Table 5: Effect of different settings on the [Inception score (IS) / Fréchet Inception Distance (FID) / Number of score Function Evaluations (NFE)] from 10k samples (with mini-batches of 1k samples) with the VE - CIFAR10 model.

| Change(s) in Algorithm 1 | IS | FID | NFE |
|---|---|---|---|
| No change $\left[q = 2, r = 0.9, \delta(\mathbf{x}', \mathbf{x}'_{prev})\right]$ | 9.39 | 4.89 | 8856 |
| Small modifications | | | |
| $\delta(\mathbf{x}')$ | 9.39 | 4.99 | 17514 |
| No Extrapolation (thus, using Euler–Maruyama) | 9.58 | 6.57 | 8802 |
| $q = \infty$ | 9.41 | 5.03 | 39500 |
| $r = 0.5$ | 9.47 | 4.87 | 9594 |
| $r = 0.8$ | 9.45 | 4.84 | 8952 |
| $r = 1$ | 9.43 | 4.93 | 8784 |
| Variations of Lamba (2003) Algorithm | | | |
| $r = 0.5$, Lamba integration | 9.08 | 18.28 | 2492 |
| $r = 0.5$, Lamba integration, Extrapolation | 3.70 | 169.78 | 2252 |
| $r = 0.5$, Lamba integration, $q = \infty$ | 9.42 | 6.80 | 5886 |
| $r = 0.5$, Lamba integration, $q = \infty$, $\theta = 0.8$ | 9.35 | 6.20 | 2970 |

We notice that using $q = \infty$ and $\delta(\mathbf{x}')$ lead to higher NFE as we expect. However, they also generally lead to higher FID, thus lower quality/diversity, which is not expected! We hypothesized that this might be due to the large number of steps taken when using $q = \infty$ and $\delta(\mathbf{x}')$. To test this, we trained the VE and VP models with Euler-Maruyama with 10k steps instead of 1k steps and we indeed obtained higher FIDs. This means that taking too many steps leads to worse performance in score-based models.

Worse quality from taking more steps should typically not happen as more steps should mean a more precise trajectory. We hypothesize this to be caused by the difference between using the actual score function instead of using the pre-trained score-network; given the errors in the score network, it may be that taking too many steps leads to some deviations from the right solution. Alternatively, this could also be due to the metric, as no existing quality/diversity metrics for generative models is truly perfect; but we believe that this hypothesis is less plausible than the increasing errors from using the score-network.

## C    DYNAMIC STEP SIZE ALGORITHM FOR SOLVING ANY TYPE OF SDE (RATHER THAN JUST REVERSE DIFFUSION PROCESSES)

Assume, we have a Diffusion Process of the form:

$$d\mathbf{x} = f(\mathbf{x}, t)dt + g(\mathbf{x}, t)d\mathbf{w}. \tag{7}$$

The algorithm to solve it is represented in Algorithm 2. The differences are:

- it is in forward-time
- the range of time must be given
- The diffusion can depend on $\mathbf{x}$, which leads to a slightly more complicated formulation that depends on some random number $s = \pm 1$ (Roberts, 2012).
- we retain the full trajectory instead of only the ending
- we retain the noise after a rejection to ensure that there is no bias in the rejections

---

**Algorithm 2** Dynamic step size extrapolation for solving arbitrary (forward-time) Diffusion Processes

---

**Require:** $s_\theta, t_{begin}, t_{end}, \epsilon_{rel}, \epsilon_{abs}, h_{init} = 0.01, r = 0.9, \theta = 0.9$

$\quad t \leftarrow t_{begin}$
$\quad h \leftarrow h_{init}$
$\quad$ Initialize $\mathbf{x}(t)$
$\quad \mathbf{x}'_{prev} \leftarrow \mathbf{x}$
$\quad$ Draw $\mathbf{z} \sim \mathcal{N}(\mathbf{0}, \mathbf{I})$
$\quad$ **while** $t < t_{end}$ **do**
$\quad\quad$ **if** Stratonovich SDE or $g(\mathbf{x}, t) = g(\mathbf{x})$ **then**
$\quad\quad\quad s \leftarrow 0$
$\quad\quad$ **else**                                                                                  ▷ Itō diffusion
$\quad\quad\quad$ Draw $s \sim \text{Uniform}(\{-1, 1\})$
$\quad\quad \mathbf{x}' \leftarrow \mathbf{x}(t) + hf(\mathbf{x}(t), t) + \sqrt{h}g(\mathbf{x}(t), t)(\mathbf{z} - s)$                ▷ Euler-Maruyama
$\quad\quad \tilde{\mathbf{x}} \leftarrow \mathbf{x}(t) + hf(\mathbf{x}', t + h) + \sqrt{h}g(\mathbf{x}', t + h)(\mathbf{z} + s)$
$\quad\quad \mathbf{x}'' \leftarrow \frac{1}{2}(\mathbf{x}' + \tilde{\mathbf{x}})$                                          ▷ Improved Euler (SDE version)
$\quad\quad \boldsymbol{\delta} \leftarrow \max(\epsilon_{abs}, \epsilon_{rel} \max(|\mathbf{x}'|, |\mathbf{x}'_{prev}|))$                  ▷ Element-wise operations
$\quad\quad E_2 \leftarrow \frac{1}{\sqrt{n}} \left\| (\mathbf{x}' - \mathbf{x}'') / \boldsymbol{\delta} \right\|_2$
$\quad\quad$ **if** $E_2 \leq 1$ **then**                                                               ▷ Accept
$\quad\quad\quad t \leftarrow t + h$
$\quad\quad\quad \mathbf{x}(t) \leftarrow \mathbf{x}''$                                                          ▷ Extrapolation
$\quad\quad\quad \mathbf{x}'_{prev} \leftarrow \mathbf{x}'$
$\quad\quad\quad$ Draw $\mathbf{z} \sim \mathcal{N}(\mathbf{0}, \mathbf{I})$
$\quad\quad h \leftarrow \min(t, \theta h E_2^{-r})$                                                    ▷ Dynamic step size update
$\quad$ **return** $\mathbf{x}$

---

## D  IMPLEMENTATION DETAILS

We started from the original code by Song et al. (2020a) but changed a few settings concerning the SDE solving. This creates some very minor difference between their reported results and ours. For the VP and VP-deep models, we obtained 2.55 and 2.49 instead of the original 2.55 and 2.41 for the baseline method (EM). For the VE and VE-deep models, we obtained 2.40 and 2.21 instead of the original 2.38 and 2.20 for the baseline method (Reverse-Diffusion with Langevin).

As done in Song et al. (2020a), we used the optimal signal-to-noise ratio of 0.01 for the Langevin corrector.

When solving the SDE, time followed the sequence $t_0 = 1$, $t_i = t_{i-1} - \frac{1-\epsilon}{N}$, where $N = 1000$ for CIFAR-10, $N = 2000$ for LSUN, $\epsilon = 1e-3$ for VP models, and $\epsilon = 1e-5$ for VE models.

Meanwhile, the actual step size $h$ used in the code for Euler-Maruyama (EM) was equal to $\frac{1}{N}$. Thus, there was a negligible difference between the step size used in the algorithm ($h = \frac{1}{N}$) and the actual step size implied by $t$ ($h = \frac{1-\epsilon}{N}$). Note that this has little to no impact.

The bigger issue is at the last predictor step was going from $t = \epsilon$ to $t = \epsilon - \frac{1}{N} < 0$. Thus, $t$ was made negative. Furthermore the sample was denoised at $t < 0$ while assuming $t = \epsilon$. There are two ways to fix this issue: 1) take only a step from $t = \epsilon$ to $t = 0$ and do not denoise (since you cannot denoise with the incorrect $t$ or with $t = 0$), or 2) stop at $t = \epsilon$ and then denoise. Since denoising is very helpful, we took approach 2; however, both approaches are sensible.

Finally, denoising was not implemented correctly before. Denoising was implemented as one predictor step (Reverse-Diffusion or EM) without adding noise. This corresponds to:

$$\mathbf{x} \leftarrow \mathbf{x} - h \left[ f(\mathbf{x}, t) - g(t)^2 \nabla_\mathbf{x} \log p_t(\mathbf{x}) \right].$$

At the last iteration, this incorrect denoising would be:

$$\mathbf{x} \leftarrow \mathbf{x} + \frac{d[\sigma^2(t)]}{dt} \frac{1}{N} \nabla_\mathbf{x} \log p_t(\mathbf{x})$$

$$= \mathbf{x} + \frac{\sigma_{min}}{N} \sqrt{2 \log \left( \frac{\sigma_{max}}{\sigma_{min}} \right)} \nabla_\mathbf{x} \log p_t(\mathbf{x})$$

$$\approx \mathbf{x}$$

for VE and

$$\mathbf{x} \leftarrow \mathbf{x} + \frac{\sqrt{\beta_{min}}}{N} \nabla_\mathbf{x} \log p_t(\mathbf{x})$$

$$\approx \mathbf{x}$$

for VP.

Meanwhile, the correct way to denoise based on Tweedie formula (Efron, 2011) is:

$$\mathbf{x} \leftarrow \mathbf{x} + \mathrm{Var}[\mathbf{x}(t)|\mathbf{x}(0)] \nabla_\mathbf{x} \log p_t(\mathbf{x}),$$

where $\mathrm{Var}[\mathbf{x}(t)|\mathbf{x}(0)]$ is the variance of the transition kernel: $\mathrm{Var}[\mathbf{x}(t)|\mathbf{x}(0)] = \sigma_{min} = 0.01$ for VE and $\mathrm{Var}[\mathbf{x}(t)|\mathbf{x}(0)] = 1$. This means that the correct Tweedie formula corresponds to

$$\mathbf{x} \leftarrow \mathbf{x} + 0.01^2 \nabla_\mathbf{x} \log p_t(\mathbf{x})$$

$$\approx \mathbf{x}$$

for VE and

$$\mathbf{x} \leftarrow \mathbf{x} + \nabla_\mathbf{x} \log p_t(\mathbf{x})$$

for VP.

As can be seen, denoising has a very small impact on VE so the difference between the correct and incorrect denoising is minor. Meanwhile, for VP the incorrect denoising lead to a tiny change, while the correct denoising lead to a large change. In practice, we observe that changing the denoising method to the correct one does not significantly affect the FID with VE, but lowers down the FID significantly with VP.

# E    Inception Score on CIFAR-10

Table 6: Inception Score on CIFAR-10 (32x32) from 50K samples

| Method | VP | VP-deep | VE | VE-deep |
|---|---|---|---|---|
| Reverse-Diffusion & Langevin | 9.94 | 9.85 | 9.86 | 9.83 |
| Euler-Maruyama | 9.71 | 9.73 | 9.49 | 9.31 |
| Ours ($\epsilon_{rel} = 0.01$) | 9.46 | 9.54 | 9.50 | 9.48 |
| Ours ($\epsilon_{rel} = 0.02$) | 9.51 | 9.48 | 9.57 | 9.50 |
| Ours ($\epsilon_{rel} = 0.05$) | 9.50 | 9.61 | 9.64 | 9.63 |
| Ours ($\epsilon_{rel} = 0.10$) | 9.69 | 9.64 | 9.87 | 9.75 |
| Probability Flow (ODE) | 9.37 | 9.33 | 9.17 | 9.32 |

# F    Stability and Bias of the Numerical Scheme

The following constructions rely on the underlying assumption of the stochastic dynamics being driven by a wiener process. More so, we also assume that the Brownian motion is time symmetrical. Both assumptions are consistent and widely used in the literature; for example, see (Gardiner, 2009) (Arnold, 1974).

The method described in Algorithm 1 gives us a significant speedup in terms of computing time and actions. Albeit the speed up comes from a piece-wise step in the algorithm combining the traditional Euler Maruyama (EM) with a form of adaptive step size predictor-corrector. Here we show that both the stability and the convergence of the EM scheme are conserved by introducing the extra adaptive stepsize of our new scheme. As a first step, we define the stability and bias in a Stochastic Differential Equation (SDE) numerical solution.

We denote $\Re(\lambda)$ as the real value of a complex-valued $\lambda$.

The linear test SDE is defined in the following way:

$$\mathrm{d}\mathbf{x}_t = \lambda \mathbf{x}_t \mathrm{d}t + \sigma \mathrm{d}\mathbf{w}_t \tag{8}$$

with its numerical counterpart

$$\mathbf{y}_{n+1} = \Re\left(h\lambda\right)\mathbf{y}_n + \mathbf{z}_n,$$

where the $\mathbf{z}_n$ are random variables that do not depend on $\mathbf{y}_0, \mathbf{y}_1......\mathbf{y}_n$ or $\lambda$ and the EM scheme is

$$\mathbf{y}_{n+1} = (1 + h\lambda)\mathbf{y}_n + \mathbf{z}_n.$$

A numerical scheme is asymptotically unbiased with step size $h > 0$ if, for a given linear SDE (8) driven by a two-sided Wiener process, the distribution of the numerical solution $\mathbf{y}_n$ converges as $n \to \infty$ to the normal distribution with zero mean and variance $\frac{\sigma^2}{2|\lambda|}$ (Artemiev and Averina, 2011). This stems from the fact that a solution of a linear SDE (8) is a Gaussian process whenever the initial condition is Gaussian (or deterministic); thus, there are only two moments that control the bias in the algorithm:

$$\lim_{n\to\infty} \mathbb{E}\left[\mathbf{y}_n\right] = 0, \qquad \lim_{n\to\infty} \mathbb{E}\left[\mathbf{y}_n^2\right] = -\frac{\sigma^2}{2\left|\lambda\right|}.$$

A numerical scheme with step size $h$ is numerically stable in mean if the numerical solution $\mathbf{y}_n^{(h)}$ applied to a linear SDE satisfies

$$\lim_{n\to\infty} \mathbb{E}\left[\mathbf{y}_n\right] = 0,$$

and is stable in mean square (Saito and Mitsui, 1996) if we have that

$$\lim_{h\to 0}\left(\lim_{n\to\infty} \mathbb{E}\left[\left|\mathbf{y}_n\right|^2\right]\right) = \frac{\sigma^2}{2\Re(\lambda)}.$$

In what follows, we will trace the criteria for bias through our algorithm and show that it remains unbiased. By construction, the first EM step remains unbiased, while for the RDP, we write down the time reverse Wiener process as

$$\tilde{\mathbf{y}}_{n+1} = (1 + \lambda h)\,\tilde{\mathbf{y}}_n + \tilde{\mathbf{z}}_n$$

in the reverse time steps $h$ i.e., $t - nh, t - 2nh$,

$$
\begin{aligned}
\mathbb{E}\left[\tilde{\mathbf{y}}_{n+1}\right] &= (1 + \lambda\,(t - h))\,\mathbb{E}\left[\tilde{\mathbf{y}}_n\right]\\
&= (1 + \lambda\,(t - h))\,\mathbb{E}\left[(1 + \lambda\,(t - h))\,\tilde{\mathbf{y}}_{n-1}\right]\\
&\quad\vdots\\
&= (1 + \lambda\,(t - h))^{n+1}\,\mathbb{E}\left[\tilde{\mathbf{y}}_0\right]\\
&= (1 + \lambda\,(t - h))^{n+1}\,\mathbb{E}\left[\mathbf{y}_0\right].
\end{aligned}
$$

Thus, if

$$|1 + \lambda\,(t - h)| < 1,$$

then

$$\lim_{n\to\infty}\mathbb{E}\left[\mathbf{y}_n^{(h)}\right] = 0.$$

In Algorithm 1, we are performing consecutive steps forward and backwards in time so $t = 2h$ such that

$$|1 + \lambda h| < 1.$$

Thus, the scheme is both numerically stable and unbiased with respect to the mean.

Next, we focus on the numerical solution in mean square:

$$
\begin{aligned}
\mathbb{E}\left[\left|\tilde{\mathbf{y}}_{n+1}\right|^2\right] &= |1 + \lambda\,(t - h)|^2\,\mathbb{E}\left[\left|\tilde{\mathbf{y}}_n\right|^2\right] + \sigma^2 h\\
&= |1 + \lambda\,(t - h)|^2\left\{|1 + \lambda\,(t - h)|^2\,\mathbb{E}\left[\left|\tilde{\mathbf{y}}_{n-1}\right|^2\right] + \sigma^2 h\right\} + \sigma^2 h\\
&\quad\vdots\\
&= |1 + \lambda\,(t - h)|^{2(n+1)}\,\mathbb{E}\left[|\mathbf{y}_0|\right] + \frac{|1 + \lambda\,(t - h)|^{2(n+1)} - 1}{2\Re\lambda + |\lambda|^2\,(t - h)}\sigma^2.
\end{aligned}
$$

Under the same assumption of consecutive steps, we have that

$$\mathbb{E}\left[\left|\tilde{\mathbf{y}}_{n+1}\right|^2\right] = |1 + \lambda h|^{2(n+1)}\,\mathbb{E}\left[|\mathbf{y}_0|\right] + \frac{|1 + \lambda h|^{2(n+1)} - 1}{2\Re(\lambda) + |\lambda|^2\,h}\sigma^2,$$

$$\lim_{n\to\infty}\mathbb{E}\left[\left|\tilde{\mathbf{y}}_{n+1}\right|^2\right] = -\frac{\sigma^2}{2\Re(\lambda) + |\lambda|^2\,h},$$

$$\lim_{h\to 0}\left(\lim_{n\to\infty}\mathbb{E}\left[\left|\tilde{\mathbf{y}}_{n+1}\right|^2\right]\right) = -\frac{\sigma^2}{2\Re(\lambda)}.$$

Assuming the imaginary part of $\lambda$ is null, we have that

$$\lim_{h\to 0}\left(\lim_{n\to\infty}\mathbb{E}\left[\left|\tilde{\mathbf{y}}_{n+1}\right|^2\right]\right) = -\frac{\sigma^2}{2\,|\lambda|}.$$

Thus, the numerical scheme is stable and unbiased in the mean square.

Following the two steps for computation of $\mathbf{x}'$ and $\tilde{\mathbf{x}}$, the step size decreases and does not change size; thus, all the above statements hold, and the entire algorithm is stable and unbiased with respect to both the mean and square mean.

# G SAMPLES

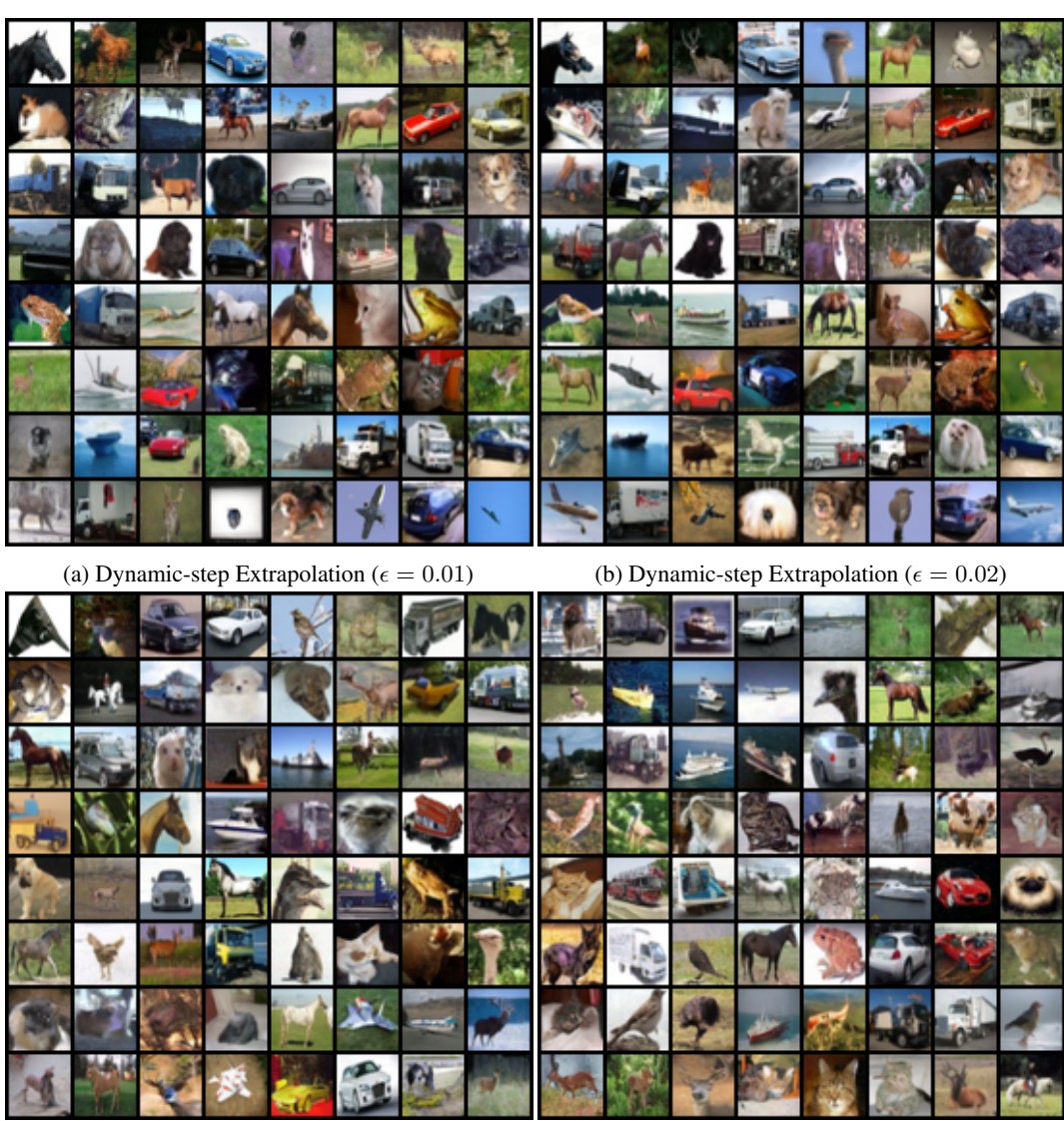

(a) Dynamic-step Extrapolation ($\epsilon = 0.01$)   (b) Dynamic-step Extrapolation ($\epsilon = 0.02$)

(c) Dynamic-step Extrapolation ($\epsilon = 0.05$)   (d) Dynamic-step Extrapolation ($\epsilon = 0.10$)

Figure 2: VP - CIFAR10

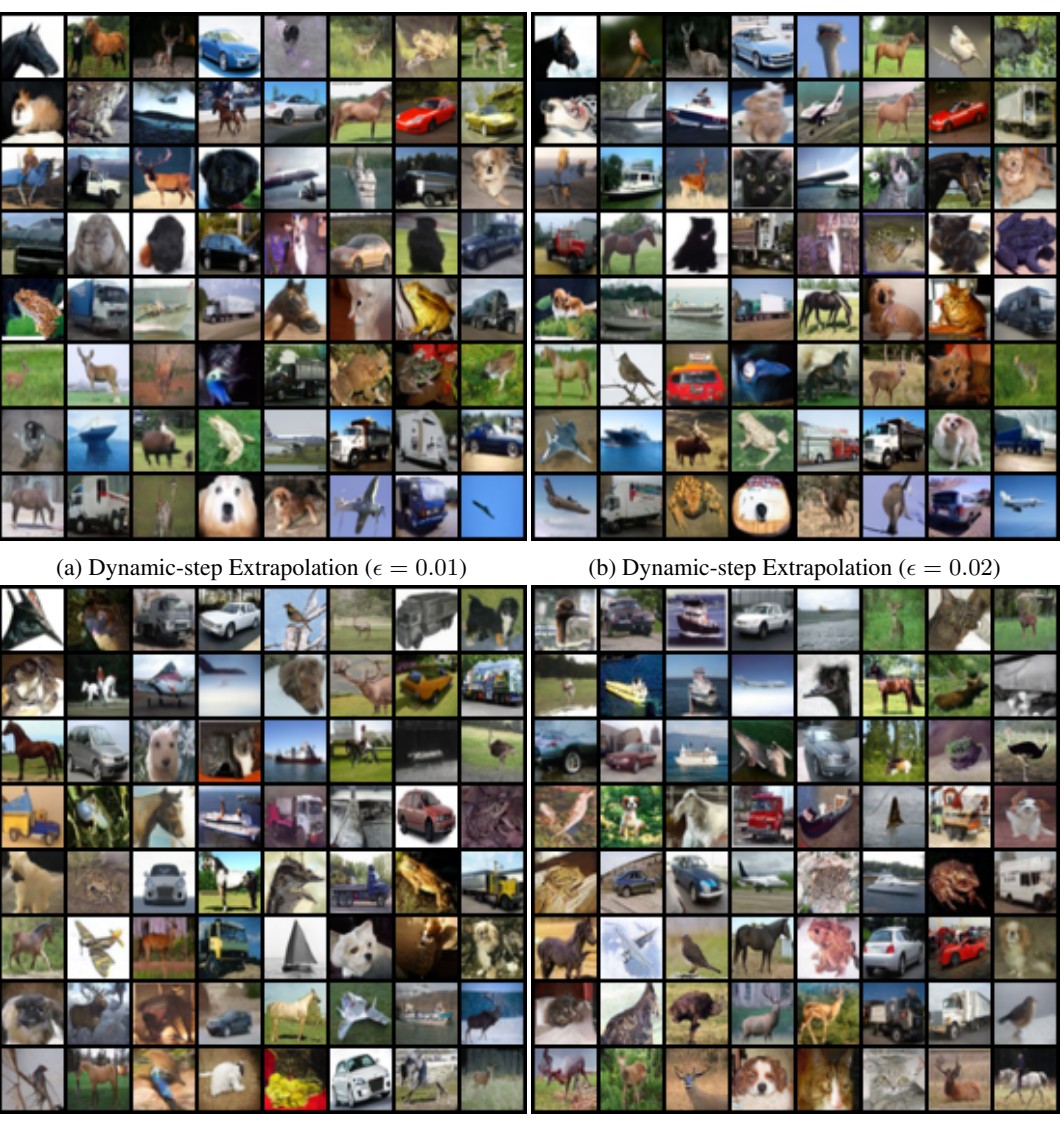

(a) Dynamic-step Extrapolation ($\epsilon = 0.01$)  (b) Dynamic-step Extrapolation ($\epsilon = 0.02$)

(c) Dynamic-step Extrapolation ($\epsilon = 0.05$)  (d) Dynamic-step Extrapolation ($\epsilon = 0.10$)

Figure 3: VP-deep - CIFAR10

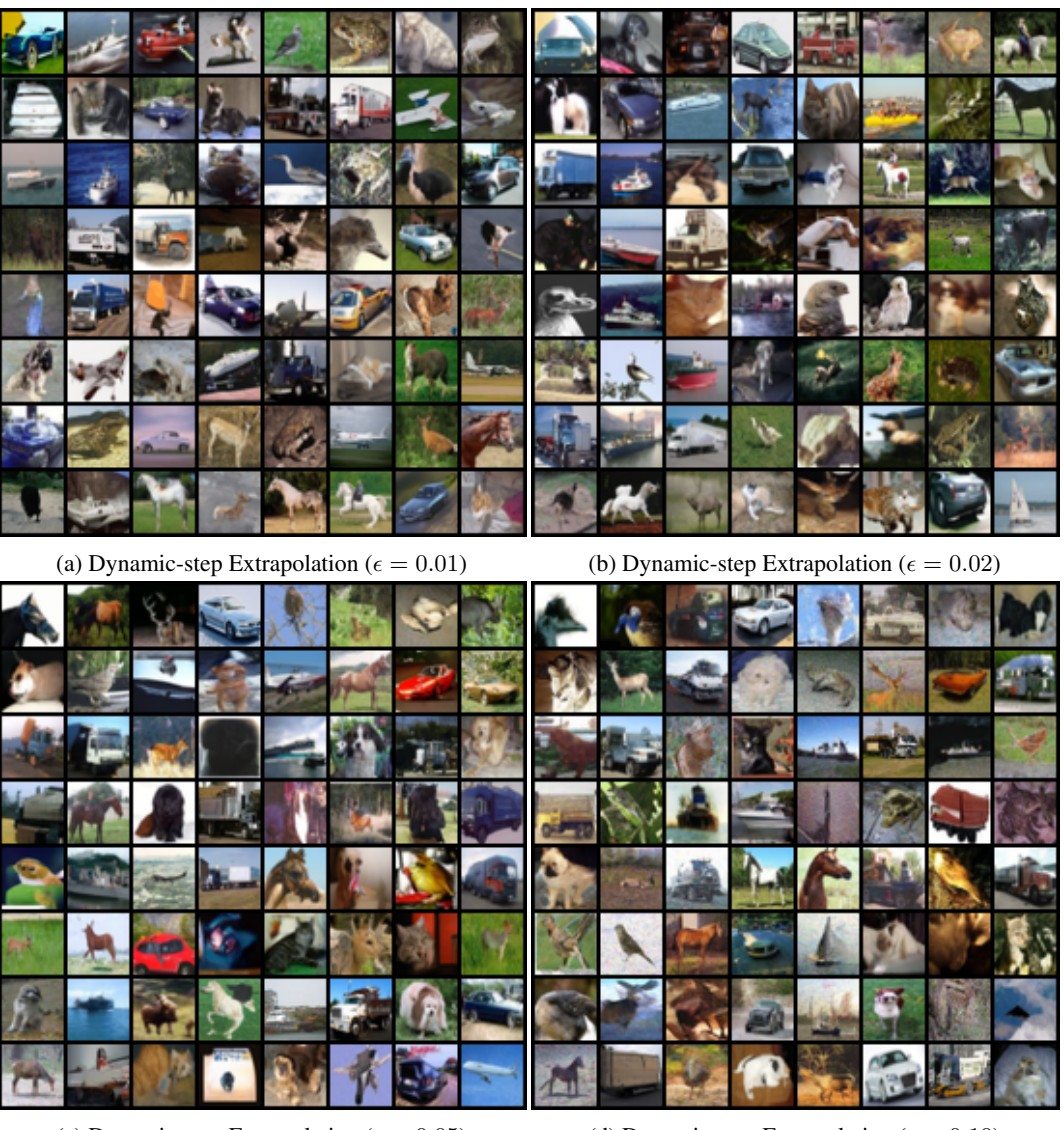

(a) Dynamic-step Extrapolation ($\epsilon = 0.01$)    (b) Dynamic-step Extrapolation ($\epsilon = 0.02$)

(c) Dynamic-step Extrapolation ($\epsilon = 0.05$)    (d) Dynamic-step Extrapolation ($\epsilon = 0.10$)

Figure 4: VE - CIFAR10

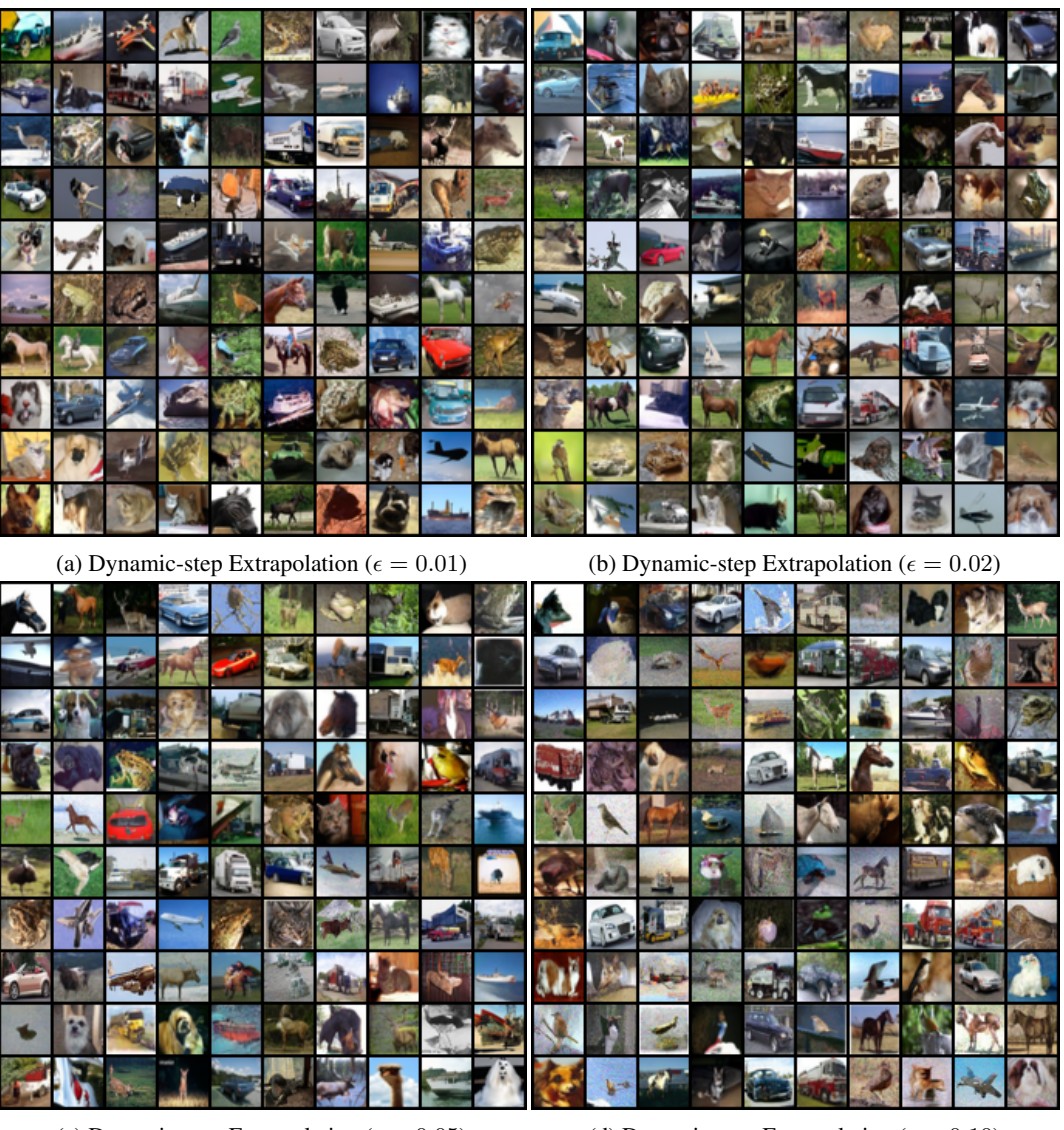

(a) Dynamic-step Extrapolation ($\epsilon = 0.01$)  (b) Dynamic-step Extrapolation ($\epsilon = 0.02$)

(c) Dynamic-step Extrapolation ($\epsilon = 0.05$)  (d) Dynamic-step Extrapolation ($\epsilon = 0.10$)

Figure 5: VE-deep - CIFAR10

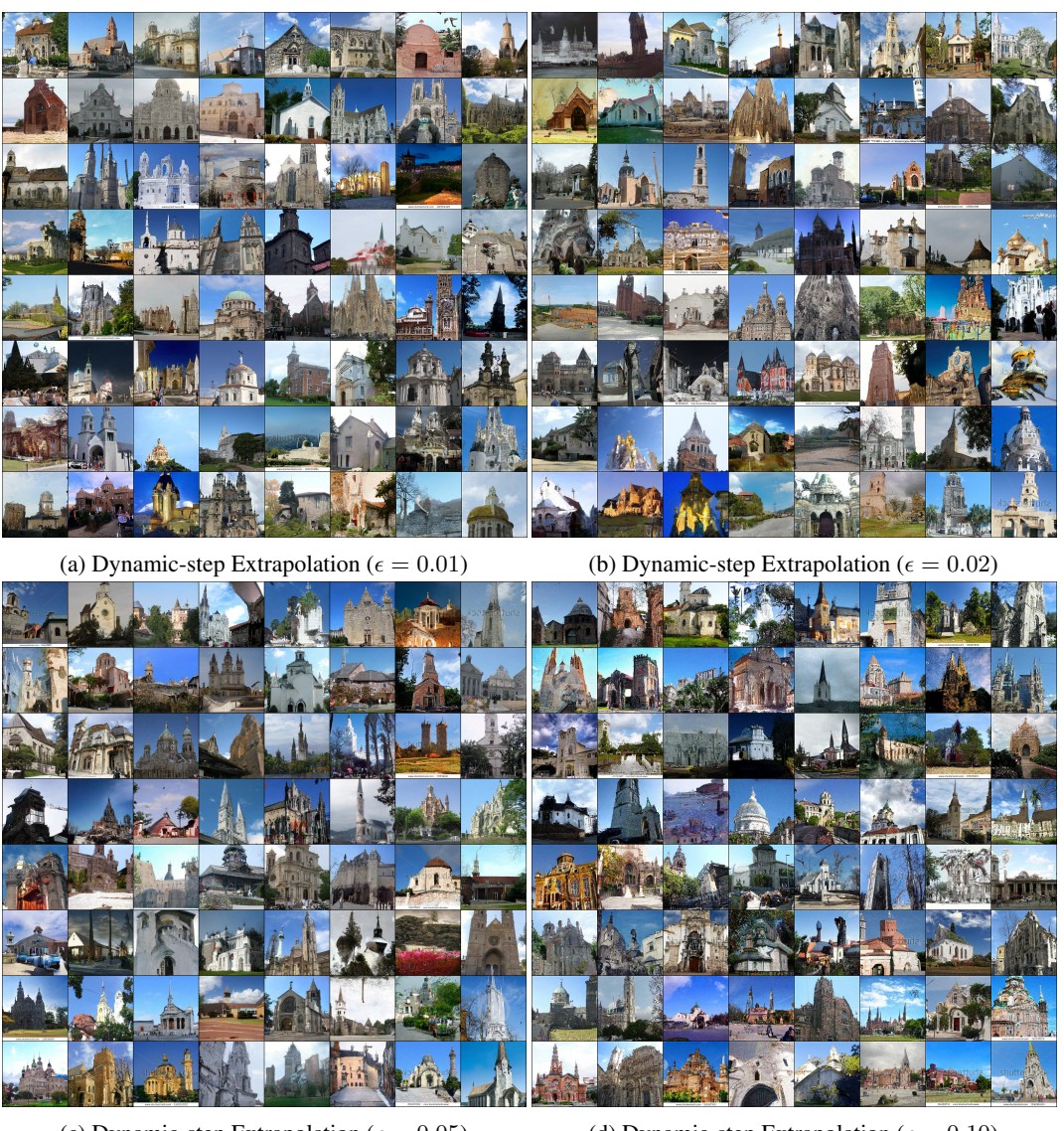

(a) Dynamic-step Extrapolation ($\epsilon = 0.01$)       (b) Dynamic-step Extrapolation ($\epsilon = 0.02$)

(c) Dynamic-step Extrapolation ($\epsilon = 0.05$)       (d) Dynamic-step Extrapolation ($\epsilon = 0.10$)

Figure 6: VE - LSUN-Church (256x256)

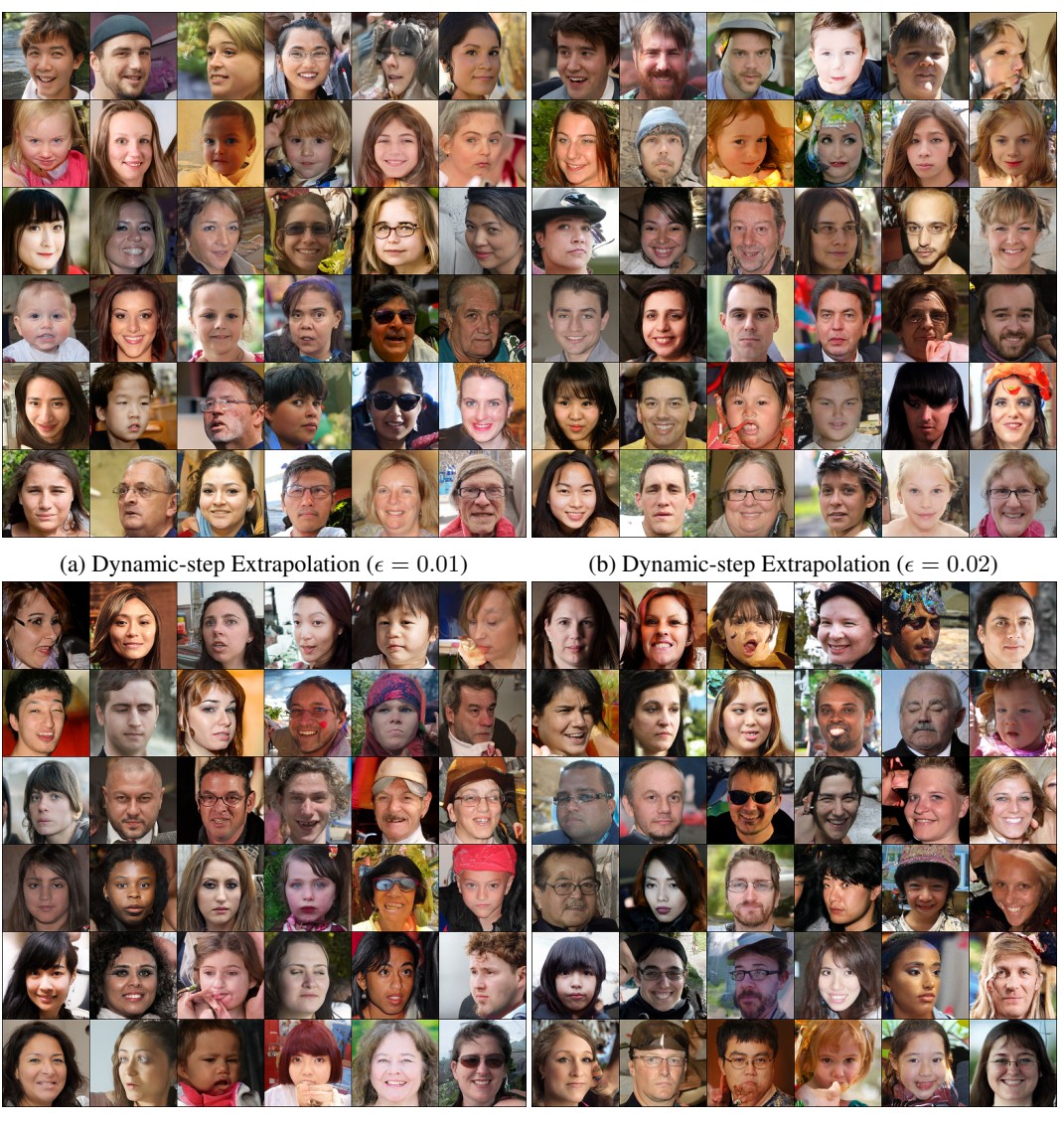

(a) Dynamic-step Extrapolation ($\epsilon = 0.01$)      (b) Dynamic-step Extrapolation ($\epsilon = 0.02$)

(c) Dynamic-step Extrapolation ($\epsilon = 0.05$)      (d) Dynamic-step Extrapolation ($\epsilon = 0.10$)

Figure 7: VE - FFHQ (256x256)

