# OpenReview forum: "Gotta Go Fast When Generating Data with Score-Based Models"
_ICLR.cc/2022/Conference — ICLR 2022 Submitted_

### Official Review · Reviewer_Px4w · 2021-11-02

**Correctness:** 3
**Technical Novelty And Significance:** 2
**Empirical Novelty And Significance:** 3
**Recommendation:** 5
**Confidence:** 3

**Main Review:**

The research problem of this paper is significant and interesting to the community of deep generative models, especially score-based models and denoising diffusion probabilistic models. The writing of this paper is clear.

Strengths:
- A novel SDE solver with a stability and bias analysis.
- The empirical success of the algorithm, i.e., high generation quality with less steps (function evaluations).
- The algorithm can readily be adapted to other score-based models with SDE.

I am not an expert in numerical solvers for differential equations, so I will leave evaluation of this part to other reviewers.

Weaknesses (and questions):
- One major concern is about the title. The title is too exaggerated and does not reflect the content of the paper. Instead, I would like to see a title that is precise, for example, "An efficient SDE solver for score-based models".
- Fig 1: What about the original Improved Euler baseline?
- Section 3: The several components of the algorithm seem to be parameter tuning and engineering trick. There is no theoretical or even intuitive justification for why they offer better performance (Appendix F does not answer this question). In addition, I do not see any ablation study on the contribution of each component.
- Section 3.1.3: The last equation ($\ell_2$ norm) in this subsection cannot be understood.
- The algorithm: Although the algorithm does not need stepsize or scheduler tuning, there are other hyperparameters involved so some tuning is still needed.
- Experiments: How often do you reject under different NFE? There should be experiments on the relationship between NFE and percentage of rejection. Next, there should be a figure similar to Fig 1 but the x-axis is the time spent rather than NFE.
- Appendix A: Are those methods $\ell_2$ or $\ell_{\infty}$?
- Appendix E: What can we read from this table?

**Summary Of The Paper:**

The paper presents a new SDE solver for the reverse process in score-based models. The algorithm is fast and offers high quality, and avoids some hyperparameter tuning. There is theoretical analysis on the stability and bias of the algorithm. The paper also has experiments comparing the proposed algorithm to several baseline methods.

**Summary Of The Review:**

The algorithm seems to be empirically successful. However, the paper lacks justification for why their algorithm can be better. There is also much space for improvements in the experiments (especially, on the overall generation speed measured by seconds). In addition, the title is exaggerated.

---------
Updates after the rebuttal

I appreciate that the authors agreed to use a more precise title. Some of my concerns are also addressed. Therefore, I will increase my score to 5. However, I still think the experiments lack some key components such as time spent and number of rejections, although they might strongly correlate to NFE. The empirical results are good but not impressive enough. I encourage the authors to further improve the method and make additional theoretical/empirical justifications.

---

> ### Author Response · Authors · 2021-11-21
> **Reviewer Px4w**
>
> > 1. One major concern is about the title. The title is too exaggerated and does not reflect the content of the paper. Instead, I would like to see a title that is precise, for example, "An efficient SDE solver for score-based models".
>
> In accordance with your suggestion, we changed the title.
>
> > 2. Fig 1: What about the original Improved Euler baseline?
>
> There is no “Improved Euler” baseline; we included all the methods (Euler, our solver, DDIM) presented in Table 1 except for “Probability Flow”. We did however test how our algorithm is affected when we change hyperparameters or the integration method (using Lamba’s integration or using no-extrapolation which means using Euler instead of Improved Euler), see Appendix B.
>
> > 3. Section 3: The several components of the algorithm seem to be parameter tuning and engineering trick. There is no theoretical or even intuitive justification for why they offer better performance (Appendix F does not answer this question). In addition, I do not see any ablation study on the contribution of each component.
>
> The skeleton of an ODE/SDE solver does not change from a paper to another; what changes are the components. We do provide a mixture of theoretical (e.g., extrapolation, improved Euler, $l_2$ norm instead of $l_\infty$ norm, our choice of absolute tolerance) and empirical justifications (tolerance involving previous $x$, choice of $r$) for choosing each component of our algorithm. There is an ablation study in Appendix B.
>
> > 4. Section 3.1.3: The last equation (ℓ2 norm) in this subsection cannot be understood.
>
> We apologize, there was a mistake. We rewrote the equation properly based on reviewer 4pA6 comment.
>
> > 5. The algorithm: Although the algorithm does not need stepsize or scheduler tuning, there are other hyperparameters involved so some tuning is still needed.
>
> We already mention this in the limitations:
>
> "Although our approach removes step size and schedule tuning, we still need to choose a value of the relative tolerance, which indirectly affects the number of steps taken; one could theoretically tune this hyper-parameter to optimize a certain metric, going against the point of removing tuning. Still, letting $\epsilon_{rel}=0.01$ for precise results and $\epsilon_{rel}=0.05$ for fast results are reasonable choices, as all evidence points to the FID being stable w.r.t. $\epsilon_{rel}$."
>
> > 6. Experiments: How often do you reject under different NFE? There should be experiments on the relationship between NFE and percentage of rejection. Next, there should be a figure similar to Fig 1 but the x-axis is the time spent rather than NFE.
>
> There are a few rejections, but they happen mostly at the beginning if the initial step-size is too big, they tend to be rare after since the score-function does not change significantly from $t$ to $t+h$ and the step-size is prevented from becoming too big through $\theta=.9$. We agree that providing the number of rejections would be interesting, but this would require us to rerun all analyses as we have never tracked the rejections consistently (outside of quick print(rejections) in some early experiments).
>
> Regarding time spent, in our early iterations of the papers (which did not have the depth of experiments that we have now), we did write down the time spent in seconds, however, we removed it due to being redundant. We say redundant because the time spent really does correlate near perfectly with the NFE and the benefit of the NFE is that it is not system dependent and thus ageless.
>
> > 7. Appendix A: Are those methods $l_2$ or $l_\infty$?
>
> $l_2$
>
> > 8. Appendix E: What can we read from this table?
>
> We observe that the Inception Score acts very strangely with score-based diffusion models, which further confirm that it is not a good measure of quality/diversity of generative models (see: https://arxiv.org/abs/1706.08500 and https://arxiv.org/abs/1801.01973). By strange, we mean the fact that a higher inception score (which means better quality/diversity) is obtained through reducing the precision of the solvers. Although not shown, we do also notice a higher inception score by taking fewer steps which means that lower precision and thus more noisy output leads to a higher inception score; this evidently should not be the case if the measure was a good metric.

---

### Official Review · Reviewer_ybV4 · 2021-11-03

**Correctness:** 3
**Technical Novelty And Significance:** 2
**Empirical Novelty And Significance:** 2
**Recommendation:** 5
**Confidence:** 3

**Main Review:**

1. The major contribution of the paper seems to be an application of existing numerical solvers (Roberts, 2012) to sampling from SDE models. The technical novelty could be limited as the solver already exists despite the tricks used to improve the empirical results.
2. There is no theoretical analysis of the method.
3. It seems that the major contribution is on the empirical side. However, the empirical results are not very impressive especially when NFE is small. For instance, in Table 1, DDIM outperforms the proposed method when NFE=49. As the goal of the paper is to improve the sampling speed of score-based models, having a good result when NFE is small enough is important. An FID of 72.29 using 49 NFE might not be impressive since NFE=49 is already a large budget. Wondering when the proposed method can outperform DDIM when NFE=100.
4. There are some formatting issues (e.g., too much space) on the first page. In the tables, some best values are in bold while some are not. It would be good to be consistent.
5. Algorithm 1 should be better explained in section 3.2. The required parameters and their selections should also be explained more clearly.
6. The paper writing can be improved. For instance, in section 3.1.1, "the stochastic Improved Euler’s method" should be explained more clearly.  "Dynamic step size algorithm" is an important contribution of the paper, the setting should be better explained. If possible, can move some details from the appendix to the main paper.
7. It would also be interesting to perform experiments on even higher-dimensional images if model checkpoints are available.

**Summary Of The Paper:**

Score-based/diffusion-based generative models can achieve high sample quality. However, their sampling speed is slow due to the large number of evaluations required by numerical SDE solvers. This works aims to accelerate the sampling process by using a more efficient SDE solver. The proposed approach generates data 2 to 10 times faster than the baselines while achieving reasonably well sample qualities.

**Summary Of The Review:**

The major contribution of the paper seems to be an application of existing numerical solvers to sampling from SDE models. The novelty could be limited as the solver already exists (despite the tricks proposed to improve empirical results). Given that the theoretical contribution is limited, the major contribution would be on the empirical side. However, the empirical results are not impressive enough: when using a small number of sampling steps, the proposed method has much worse results compared to DDIM. As the goal of the paper is to efficiently sample from SDE models, having strong performance when the sampling steps are small is important.

The paper writing can be improved. There are some formatting issues.

---

> ### Author Response · Authors · 2021-11-21
> **Reviewer ybV4**
>
> > 1. The major contribution of the paper seems to be an application of existing numerical solvers (Roberts, 2012) to sampling from SDE models. The technical novelty could be limited as the solver already exists despite the tricks used to improve the empirical results.
>
> We would like to emphasize that the novelty here is the combination of different tools which furthermore come from communities outside the ML community. We are not simply using ML tools to solve ML problems, but using tools from a very small mathematical community. Very few SDE solvers have been devised and most of them have been published in mathematical papers in the last 10 years. These tools are unknown to the large majority of the ML community; bringing those tools to the surface and combining them in the right way to solve an ML problem is novelty.
>
> Furthermore, we would like to emphasize that most adaptive step-size ODE and SDE solvers have the same structure, the main differences are the integration method and the hyperparameters. We thus carefully chose our integration method and hyperparameters based on a mixture of theory and experiments. Note that the integration method we chose by Roberts (2012) does not use adaptive step-sizes and our specific adaptive step-size algorithm with extrapolation is novel.
>
> > 2. There is no theoretical analysis of the method.
>
> We would like to quickly point out that we have stability analyses of Euler-Maruyama in an adaptive step-size scheme in Appendix F.
>
> Also importantly, the integration method (improved Euler for SDEs) has already been analyzed theoretically by Roberts (2012). We now mention in the text that Euler-Maruyama has strong-order 0.5 and Improved Euler has strong-order 1.
>
> > 3. The empirical results are not very impressive especially when NFE is small. In Table 1, DDIM outperforms the proposed method when NFE=49. As the goal of the paper is to improve the sampling speed of score-based models, having a good result when NFE is small enough is important. An FID of 72.29 using 49 NFE might not be impressive since NFE=49 is already a large budget. Wondering when the proposed method can outperform DDIM when NFE=100.
>
> In our models, DDIM (and all other methods) performed poorly with small NFEs (as can be seen in Table 1). This could be due to differences between Song et al. (2020a) continuous-time score-matching and the DDIM training procedure and architecture based on Ho et al. (2020). This is already mentioned in section 4.3.
>
> We agree that making the NEF smaller than 100 is a worthy goal. Understanding and testing why Song et al. (2020) models are slower than Ho et al. (2020) models could help us achieve that goal, but this is not the goal of the paper. For the current paper, we show how to improve the speed of Song et al. (2020) models and that we do better than DDIM except when N<=100 in which case all methods perform badly.
>
>
> > 4. There are some formatting issues (e.g., too much space) on the first page. In the tables, some best values are in bold while some are not. It would be good to be consistent.
>
> We fixed the spacing issues on the first page; it was due to the Figure on page 2. Thank you for noticing this!
> Regarding the bolding, we bold the best two values for each setting. We bold two values instead of one because sometimes the FID difference between the best and second-best models is very small.
>
> > 5. Algorithm 1 should be better explained in section 3.2. The required parameters and their selections should also be explained more clearly.
>
> To aid the reader, in section 3.2, we added a quick recap on what constitutes Algorithm 1 based on the choices made in section 3.1.
>
> > 6. The paper writing can be improved. For instance, in section 3.1.1, "the stochastic Improved Euler’s method" should be explained more clearly. "Dynamic step size algorithm" is an important contribution of the paper, the setting should be better explained. If possible, can move some details from the appendix to the main paper.
>
> We changed the notation from stochastic and non-stochastic Improved Euler to ‘Improved Euler ODE solver’ and ‘Improved Euler SDE solver’ and added some extra clarity to make clear that both methods are similar but one is for ODE and the other is a more recent variant for solving SDEs.
>
> Regarding the “dynamic step size”, we added more clarity to the concept on p5. The key idea is to estimate the error so that one can move faster when the error is thought to be small. Meanwhile, a fixed-step algorithm, even if fully tuned to perfection, will have to move at a fixed speed even when it reaches a region with little noise and thus little error.
>
> > 7. It would also be interesting to perform experiments on even higher-dimensional images if model checkpoints are available.
>
> We experimented with 256x256 images in two different datasets (FFHQ and Church), which is already high-resolution. There is a checkpoint for 1024x1024 images, but this would require more compute than we have to run.

---

### Official Review · Reviewer_RPn6 · 2021-11-04

**Correctness:** 2
**Technical Novelty And Significance:** 2
**Empirical Novelty And Significance:** 3
**Recommendation:** 6
**Confidence:** 3

**Details Of Ethics Concerns:**

There is no need to consider the ethical concern regarding on this submission

**Main Review:**

1)
I understand the paper's nature lies in the empirical side. However, some claims can be further elaborated through formal analyses. For example, Norm calculation in 3.1.3 can be analyzed by following algorithmic complexity, i.e. big-oh notations. I suggest that authors find such analysis opportunity throughout the paper. Moreover, it would be great if you can come up with a table to organize such complexity comparisons.

2)
I cannot follow why Eq 5 could be better in both quality and speed at the same time. Eq 5 has more constraints to setup the tolerance, so the speed-up seems to be an obvious benefit. However, the quality could be damaged while I also note that the quality is not being too much hurt from your result report. Why is that?

This question goes same to utilizing L2 norm, instead of L-$\infty$ norm

3)
Is there any theoretic argument on the integration method? Which can be proved or contests as a proposition or a theorem?
I could not fully comprehend the argument in the current text, and it just looks like swapping SDE solvers to find the best match.

4)
These improvements can be further investigated through ablation studies, which seems to be must-do, in my opinion. Without the ablation study, I cannot argue which technique contributed to more or less.

**Summary Of The Paper:**

This paper presents a number of treatments to speed up and improve the generation quality in the process of the reverse diffusion process, diffusion-based generative models. This paper is mostly empirical by nature, and authors suggested five techniques in improving the SDE solver performance.

**Summary Of The Review:**

Good paper with interesting and essential ideas in improving the usability of diffusion models. However, the current experimental result and the method justification are weak.

---

> ### Author Response · Authors · 2021-11-21
> **Reviewer RPn6**
>
> > 1) some claims can be further elaborated through formal analyses. Norm calculation in 3.1.3 can be analyzed by following algorithmic complexity, i.e. big-oh notations. I suggest that authors find such analysis opportunity throughout the paper. Moreover, it would be great if you can come up with a table to organize such complexity comparisons.
>
> We would like to quickly point out that we have stability analyses of Euler-Maruyama in an adaptive step-size scheme in Appendix F.
> Regarding algorithm complexity with big-oh notation, in SDE/ODE literature, it is generally used to determine the weak-order and strong-order of the solver. An algorithm has strong-order $p$ when the local error from $t$ to $t+h$ is $\mathcal{O}(h^{p+1})$. We now mention this mathematical definition on p7.
>
> The Improved Euler solver has strong-order 1, while the Euler-Maruyama solver has strong-order 0.5. This has already been proven by Roberts (https://arxiv.org/abs/1210.0933). We now make this information clear on p7.
>
> Regarding comparing the different complexities, we already do so in the table of Appendix A (which includes most methods from the DifferentialEquations.jl package). Our method being strong-order 1 is a balance between low-order (fast) methods of strong-order 0.5 and high-order (slow) methods of strong-order 1.5 we find in the DifferentialEquations.jl package. Of note, we do not mention the weak-order of the methods as convergence in expectation is not relevant for generative modeling.
>
> > 2) Why Eq 5 could be better in both quality and speed at the same time? This question goes same to utilizing L2 norm, instead of L-∞norm
>
> The speedup with the tolerance in eq 5 is because it is greater or equal to the tolerance in eq 4, which means a smaller error $E_q$. For L-∞ norm, as mentioned in the text, it is bound to lead to higher errors because a single pixel in the 3 x height x width total pixels of an image with high error will cause the L-∞ norm to be large and thus the error $E_q$ to be large.
>
> Regarding quality, there appears to be a pattern that excessively long generation processes with many steps (10k or more) lead to higher FID (thus lower quality images). We tested that theory in simple VE and VP models, with Euler-Maruyama running over 10k steps instead of the default 1k steps. We indeed found that training for too many steps leads to worse performance (higher FID), which explains why quality drops with the tolerance from eq 5 and the  L-∞norm as these methods increase the number of steps taken.
>
> Worse quality from taking more steps should typically not happen as more steps should mean a more precise trajectory. We hypothesize this to be caused by the difference between using the actual score function instead of using the pre-trained score-network; given the errors in the score network, it may be that taking too many steps leads to some deviations from the right solution. We now mention this in the Appendix, thank you for pointing this out.
>
> > 3) Is there any theoretic argument on the integration method? Which can be proved or contests as a proposition or a theorem? I could not fully comprehend the argument in the current text, and it just looks like swapping SDE solvers to find the best match.
>
> There is less theory for choosing methods and hyperparameters with SDEs in the context of generative models due to the complexity involved and the fact that different SDEs have different noise levels at different times, which affect speed and convergence. We chose improved Euler because it brings us from strong-order 0.5 to strong-order 1 through only one extra function evaluation per step (2 instead of 1), while methods of strong-order 1.5 or higher can require more. With that being said, our choice was inspired and chosen by methodologies in the numerical stochastic differential equation. It is known that for time-reversible stochastic processes such as Brownian motion, Wiener processes, and even some aspects of colored noise, Euler and higher-order symplectic schemes have convergences bound and respect/conserve norm with any underlying solution. For example, see: [1](https://www.sciencedirect.com/science/article/pii/S0304414902001503), [2](https://users.aalto.fi/~asolin/sde-book/sde-book.pdf), and [3](https://academic.oup.com/imajna/article-abstract/27/3/479/744803).
>
> Having this property that for a given time-reversible/Markovian process Euler based schemes and adaptive schemes have theoretical bounds that ensure fidelity to the theoretical solution is part of our theoretical reasoning and inspiration to choose such methods
>
> > 4) These improvements can be further investigated through ablation studies, which seems to be must-do, in my opinion. Without the ablation study, I cannot argue which technique contributed to more or less.
>
> We already did an ablation study in Appendix B where we show the effect of changing any part of the algorithm (tolerance type, integration method, q, r).

---

> > ### Comment · Reviewer_RPn6 · 2021-11-25
> > **Thanks**
> >
> > Authors responded my questions in details. I updated my score to be on the accept side.
> > Thanks

---

### Official Review · Reviewer_4pA6 · 2021-11-08

**Correctness:** 3
**Technical Novelty And Significance:** 4
**Empirical Novelty And Significance:** 3
**Recommendation:** 8
**Confidence:** 4

**Main Review:**

## Strengths

1. This paper proposes the first adaptive step-size numerical SDE solver for score-based generative modeling. Experimental results demonstrate clear improvement over previous methods like Euler-Maruyama methods and probability flow ODEs. It also demonstrates improvement over DDIM for a moderate budget of iteration numbers.

2. The method itself is simple and clear, and only requires tuning one hyperparameter.

## Weaknesses

1. No theoretical understanding of the proposed numerical SDE solver. It would be better to include an analyze on the convergence order of the proposed approach.

2. Results of predictor-corrector for VP SDEs in Table 1 are significantly worse than results reported in the original score sde paper. What could be the reason for this? Did you use the correct signal-to-noise ratio for the Langevin corrector?

3. Some minor writing issues. The first page has a footnote "equal contribution", while all author names should be anonymized. In section 3.1.3, $E_q$ is defined to be a scalar (the $L_q$ norm), but it is referred as a vector in the expression for computing $\| x\|_2 $.

4. In section 3.1.5, it is mentioned that a different step size is applied to each data sample. How is this implemented in deep learning frameworks? Will it cause noticeable slowdown on GPUs?

**Summary Of The Paper:**

This paper proposes a new numerical solver for stochastic differential equations and demonstrated significant improvement over existing ones such as the Euler-Maruyama solver in terms of the quality/computation trade-off for score-based generative modeling with SDEs.

**Summary Of The Review:**

This paper proposes a simple and effective numerical SDE integrator for score-based generative modeling based on SDEs. Paper can be stronger with a deeper theoretical understanding.

---

> ### Author Response · Authors · 2021-11-21
> **Reviewer 4pA6**
>
> > 1. No theoretical understanding of the proposed numerical SDE solver. It would be better to include an analysis on the convergence order of the proposed approach.
>
> We provide a stability analysis of Euler-Maruyama in an adaptive step-size scheme in Appendix F. Regarding convergence order, it is known that Euler-Maruayma is strong-order 0.5 and it has been shown that Improved Euler is strong-order 1 (proved in https://arxiv.org/abs/1210.0933). We now mention this on page 7.
>
> > 2. Results of predictor-corrector for VP SDEs in Table 1 are significantly worse than results reported in the original score sde paper. What could be the reason for this? Did you use the correct signal-to-noise ratio for the Langevin corrector?
>
> Thank you for noticing this, we were not aware that we were using the incorrect snr for VP models. We were using the default SNR from the original code of the score-sde paper (SNR = .16 as the config from the original code does not change that default, see: https://github.com/yang-song/score_sde/blob/main/configs/vp/cifar10_ncsnpp_deep_continuous.py), but from the paper, it appears that the correct SNR is .01. We reran the predictor-corrector models (reverse-diffusion with Langevin) with the correct snr and updated the results in Table 1. We now obtain 3.41 and 3.28 for VP and VP-deep respectively instead of 4.27 and 4.69.
>
> > 3. Some minor writing issues.
>
> We removed the footnote and fixed the issue with $E_q$.
>
> > 4. In section 3.1.5, it is mentioned that a different step size is applied to each data sample. How is this implemented in deep learning frameworks? Will it cause noticeable slowdown on GPUs?
>
> Rather than having $t$ be a scalar, we make $t$ a vector of the same length as the batch size. Thus, every element of the mini-batch has a different $t$ and future step-size $h$ (to go from $t$ to $t+h$). Since the score-network already takes as input a vector $t$ and considering that all operations are done in a vectorized fashion, there is no slowdown on GPUs.

---

> > ### Comment · Reviewer_4pA6 · 2021-11-30
> > **Thanks for the response**
> >
> > I would like to thank the authors for their response and I am happy to keep my original rating.

---

### Decision · Program_Chairs · 2022-01-20

**Decision:**

Reject

**Comment:**

The paper proposes numerical method for solving SDEs that empirically are faster than previous approaches. Two reviewers felt the paper was above threshold, while two felt it was below threshold for acceptance. While the paper is borderline in this sense, all four reviewers noted that the paper lacked a theoretical justification and rested on empirical evidence for the usefulness of the approach. Several reviewers also pointed out that these empirical results are on the weak side. While the paper may add a potentially useful learning trick to the optimization literature, these two significant concerns put it on the side of a borderline reject.